# Bridging OOD Detection and Generalization:
# A Graph-Theoretic View

**Han Wang**[*]
Department of Electrical and Computer Engineering
University of Illinois Urbana-Champaign
hanw14@illinois.edu

**Yixuan Li**
Department of Computer Sciences
University of Wisconsin-Madison
sharonli@cs.wisc.edu

## Abstract

In the context of modern machine learning, models deployed in real-world scenarios often encounter diverse data shifts like covariate and semantic shifts, leading to challenges in both out-of-distribution (OOD) generalization and detection. Despite considerable attention to these issues separately, a unified framework for theoretical understanding and practical usage is lacking. To bridge the gap, we introduce a graph-theoretic framework to jointly tackle both OOD generalization and detection problems. By leveraging the graph formulation, data representations are obtained through the factorization of the graph's adjacency matrix, enabling us to derive provable error quantifying OOD generalization and detection performance. Empirical results showcase competitive performance in comparison to existing methods, thereby validating our theoretical underpinnings. Code is publicly available at https://github.com/deeplearning-wisc/graph-spectral-ood.

## 1 Introduction

Machine learning models deployed in real-world applications often confront data that deviates from the training distribution in unforeseen ways. As depicted in Figure 1, a model trained on in-distribution (ID) data (e.g., seabirds) may encounter data exhibiting *covariate shifts*, such as birds in forest environments. In this scenario, the model must retain its ability to accurately classify these covariate-shifted out-of-distribution (OOD) samples as birds—an essential capability known as OOD generalization [1, 2]. Alternatively, the model may encounter data with novel semantics, like dogs, which it has not seen during training. In this case, the model must recognize these *semantic-shifted* OOD samples and abstain from making incorrect predictions, underscoring the significance of OOD detection [3, 4]. Thus, for a model to be considered robust and reliable, it must excel in both OOD generalization and detection, tasks that are often addressed separately in current research.

Recently, Bai et al. [5] introduced a framework that addresses both OOD generalization and detection simultaneously. The problem setting leverages unlabeled wild data naturally arising in the model's operational environment, representing it as a composite distribution of ID, covariate-shifted OOD, and semantic-shifted OOD data. While such data is ubiquitously available in many real-world applications, harnessing the power of wild data is challenging due to the heterogeneity of the wild data distribution—the learner lacks clear membership (ID, Covariate-OOD, Semantic-OOD) for samples drawn from the wild data distribution. Despite empirical progress made, *a formalized understanding of how wild data impacts OOD generalization and detection is still lacking*.

In this paper, we formalize a graph-theoretic framework for understanding OOD generalization and detection problems jointly. We begin by formulating a graph, where the vertices are all the data points and edges connect similar data points. These edges are defined based on a combination of supervised

---

[*]Work done while visiting UW-Madison.

38th Conference on Neural Information Processing Systems (NeurIPS 2024).

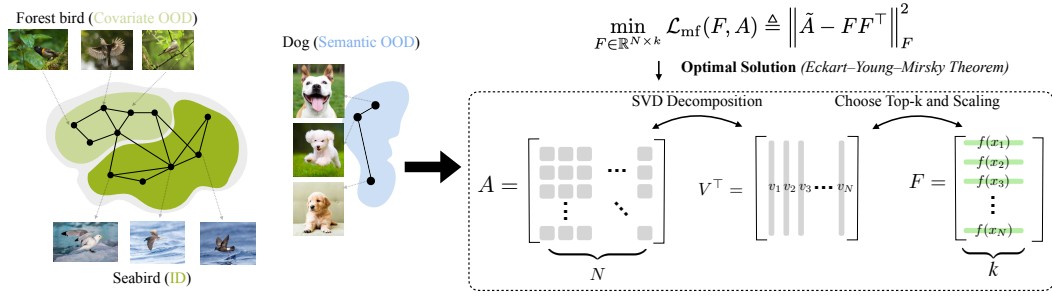

Figure 1: Illustration of our graph-theoretic framework for joint out-of-distribution generalization and detection. **Left**: Graph formulation containing three types of data in the wild: ID (e.g., seabird), covariate OOD (e.g., bird in the forest), and semantic OOD (e.g., dog). **Right**: Graph factorization for obtaining the closed-form solution of the data representations, which are used to derive OOD generalization and OOD detection errors.

and self-supervised signals, incorporating both labeled ID data and unlabeled wild data. By modeling the connectivity among data points, we can uncover meaningful sub-structures in the graph (e.g., covariate-shifted OOD data is embedded closely to the ID data, whereas semantic-shifted OOD data is distinguishable from ID data). Importantly, this graph serves as a foundation for understanding the impact of wild unlabeled data on both OOD generalization and detection, enabling a theoretical characterization of performance through graph factorization. Within this framework, we derive a formal linear probing error, quantifying the misclassification rate on covariate-shifted OOD data. Furthermore, our framework yields a closed-form solution that quantifies the distance between ID and semantic OOD data, directly elucidating OOD detection performance (Section 4).

Beyond theoretical analysis, our graph-theoretic framework can be used practically. In particular, the spectral decomposition can be equivalently achieved by minimizing a surrogate objective, which can be efficiently optimized end-to-end using modern neural networks. Thus, our approach enjoys theoretical guarantees while being applicable to real-world data. Experimental results demonstrate the effectiveness of our graph-based approach, showcasing substantial improvements in both OOD generalization and detection performance. In comparison to the state-of-the-art method Scone [5], our approach achieves a significant reduction in FPR95 by an average of 8.34% across five semantic-shift OOD datasets (Section 5). We summarize our main contributions below:

1. We introduce a graph-theoretic framework for understanding both OOD generalization and detection, formalizing it by spectral decomposition of the graph containing ID, covariate-shift OOD data, and semantic-shift OOD data.

2. We provide theoretical insights by quantifying OOD generalization and detection performance through provable error, based on the closed-form representations derived from the spectral decomposition on the graph.

3. We evaluate our model's performance through a comprehensive set of experiments, providing empirical evidence of its robustness and its alignment with our theoretical analysis. Our model consistently demonstrates strong OOD generalization and OOD detection capabilities, achieving competitive results when benchmarked against the existing state-of-the-art.

## 2 Problem Setup

We consider the empirical training set $\mathcal{D}_l \cup \mathcal{D}_u$ as a union of labeled and unlabeled data. The labeled set $\mathcal{D}_l = \{\bar{x}_i, y_i\}_{i=1}^n$, where $y_i$ belongs to *known* class space $\mathcal{Y}_l$. Let $\mathbb{P}_{in}$ denote the marginal distribution over input space, which is referred to as the in-distribution (ID). Following Bai et al. [5], the unlabeled set $\mathcal{D}_u = \{\bar{x}_i\}_{i=1}^m$ consists of ID, covariate OOD, and semantic OOD data, where each sample $\bar{x}_i$ is drawn from a mixture distribution defined below.

**Definition 2.1.** *The marginal distribution of the wild data is defined as:*

$$\mathbb{P}_{wild} := (1 - \pi_c - \pi_s)\mathbb{P}_{in} + \pi_c \mathbb{P}_{out}^{covariate} + \pi_s \mathbb{P}_{out}^{semantic},$$

*where $\pi_c, \pi_s, \pi_c + \pi_s \in [0, 1]$. $\mathbb{P}_{in}$, $\mathbb{P}_{out}^{covariate}$, and $\mathbb{P}_{out}^{semantic}$ represent the marginal distributions of ID, covariate-shifted OOD, and semantic-shifted OOD data respectively.*

**Learning goal.** We aim to learn jointly an OOD detector $g_\theta \colon \mathcal{X} \to \{\text{IN}, \text{OUT}\}$ and a multi-class classifier $f_\theta$, by leveraging labeled ID data $\mathcal{D}_l$ and unlabeled wild data $\mathcal{D}_u$. Let $\hat{y}(f_\theta(\bar{x})) := \arg\max_y f_\theta^{(y)}(\bar{x})$, where $f_\theta^{(y)}(\bar{x})$ denotes the $y$-th element of $f_\theta(\bar{x})$, corresponding to label $y$. We notate $g_\theta$ and $f_\theta$ with parameters $\theta$ to indicate that these functions share neural network parameters. In our model evaluation, we are interested in three metrics:

**Definition 2.2.** *We define ID generalization accuracy (ID-Acc), OOD generalization accuracy (OOD-Acc), and OOD detection error as follows:*

$$\uparrow \text{ID-Acc}(f_\theta) := \mathbb{E}_{(\bar{x},y) \sim \mathbb{P}_{in}}(\mathbb{1}\{\hat{y}(f_\theta(\bar{x})) = y\}),$$

$$\uparrow \text{OOD-Acc}(f_\theta) := \mathbb{E}_{(\bar{x},y) \sim \mathbb{P}_{out}^{covariate}}(\mathbb{1}\{\hat{y}(f_\theta(\bar{x})) = y\}),$$

$$\downarrow \text{FPR}(g_\theta) := \mathbb{E}_{\bar{x} \sim \mathbb{P}_{out}^{semantic}}(\mathbb{1}\{g_\theta(\bar{x}) = \text{IN}\}),$$

*where $\mathbb{1}\{\cdot\}$ represents the indicator function, and the arrows indicate the directionality of improvement (higher/lower is better). For OOD detection, ID samples are considered positive and FPR signifies the false positive rate.*

## 3 Graph-Based Framework for OOD Generalization and Detection

### 3.1 Graph Formulation

We start by formally defining the graph and adjacency matrix. We use $\bar{x}$ to denote the set of all natural data (raw inputs without augmentation). Given an $\bar{x}$, we use $\mathcal{T}(x|\bar{x})$ to denote the probability of $x$ being augmented from $\bar{x}$, and $\mathcal{T}(\cdot|\bar{x})$ to denote the distribution of its augmentation. For instance, when $\bar{x}$ represents an image, $\mathcal{T}(\cdot|\bar{x})$ can be the distribution of common augmentations [6] such as Gaussian blur, color distortion, and random cropping. We define $\mathcal{X}$ as a general population space, which contains the set of all augmented data. In our case, $\mathcal{X}$ is composed of augmented samples from both labeled ID data $\mathcal{X}_l$ and unlabeled wild data $\mathcal{X}_u$, with cardinality $|\mathcal{X}| = N$.

We define the graph $G(\mathcal{X}, w)$ with vertex set $\mathcal{X}$ and edge weights $w$. Given our data setup, edge weights $w$ can be decomposed into two components: (1) *self-supervised connectivity* $w^{(u)}$ by treating all points in $\mathcal{X}$ as entirely unlabeled, and (2) *supervised connectivity* $w^{(l)}$ by incorporating labeled information from $\mathcal{X}_l$ to the graph. We define the connectivity formally below.

**Definition 3.1** (Self-supervised connectivity). *For any two augmented data $x, x' \in \mathcal{X}$, $w_{xx'}^{(u)}$ denotes the marginal probability of generating the positive pair [7]:*

$$w_{xx'}^{(u)} \triangleq \mathbb{E}_{\bar{x} \sim \mathbb{P}} \mathcal{T}(x|\bar{x}) \mathcal{T}(x'|\bar{x}), \tag{1}$$

*where $x$ and $x'$ are augmented from the same image $\bar{x} \sim \mathbb{P}$, and $\mathbb{P}$ is the marginal distribution of both labeled and unlabeled data. A larger $w_{xx'}^{(u)}$ indicates stronger similarity between $x$ and $x'$.*

Moreover, when having access to the labeling information for ID data, we can define the edge weight by adding additional supervised connectivity to the graph. We consider $(x, x')$ a positive pair when $x$ and $x'$ are augmented from two labeled samples $\bar{x}_l$ and $\bar{x}'_l$ with the same known class $i \in \mathcal{Y}_l$. The total edge connectivity can be formulated as below:

**Definition 3.2** (Total edge connectivity). *Considering both self-supervised and supervised connectivities, the overall similarity for any pair of data $(x, x')$ is formulated as:*

$$w_{xx'} = \eta_u w_{xx'}^{(u)} + \eta_l w_{xx'}^{(l)}, \text{where } w_{xx'}^{(l)} \triangleq \sum_{i \in \mathcal{Y}_l} \mathbb{E}_{\bar{x}_l \sim \mathbb{P}_{l_i}} \mathbb{E}_{\bar{x}'_l \sim \mathbb{P}_{l_i}} \mathcal{T}(x|\bar{x}_l) \mathcal{T}(x'|\bar{x}'_l), \tag{2}$$

*where $\mathbb{P}_{l_i}$ is the distribution of labeled samples with class label $i \in \mathcal{Y}_l$, and the coefficients $\eta_u, \eta_l$ modulate the relative importance between the two terms.*

**Adjacency matrix.** Having established the notion of connectivity, we can define the adjacency matrix $A \in \mathbb{R}^{N \times N}$ with entries $A_{xx'} = w_{xx'}$. The adjacency matrix can be decomposed into the summation of self-supervised adjacency matrix $A^{(u)}$ and supervised adjacency matrix $A^{(l)}$:

$$A = \eta_u A^{(u)} + \eta_l A^{(l)}. \tag{3}$$

As a standard technique in graph theory [8], we use the *normalized adjacency matrix*:

$$\tilde{A} \triangleq D^{-\frac{1}{2}} A D^{-\frac{1}{2}}, \tag{4}$$

where $D \in \mathbb{R}^{N \times N}$ is a diagonal matrix with $D_{xx} = w_x = \sum_{x' \in \mathcal{X}} w_{xx'}$, indicating the total edge weights connected to a vertex $x$. The normalized adjacency matrix defines the probability of $x$ and $x'$ being considered as the positive pair. The normalized adjacency matrix allows us to perform spectral decomposition as we show next.

## 3.2 Learning Representations Based on Graph Spectral

In this section, we perform spectral decomposition or spectral clustering [9]—a classical approach to graph partitioning—to the adjacency matrices defined above. This process forms a matrix where the top-$k$ eigenvectors are the columns and *each row of the matrix can be viewed as a $k$-dimensional representation of an example*. The resulting feature representations enable us to rigorously analyze the separability of ID data from semantic OOD data in a closed form, as well as the generalizability to covariate-shifted OOD data (more in Section 4).

Towards this end, we consider the following optimization, which performs low-rank matrix approximation on the adjacency matrix:

$$\min_{F \in \mathbb{R}^{N \times k}} \mathcal{L}_{\mathrm{mf}}(F, A) \triangleq \left\| \tilde{A} - FF^\top \right\|_F^2, \tag{5}$$

where $\| \cdot \|_F$ denotes the matrix Frobenious norm. According to the Eckart–Young–Mirsky theorem [10], the minimizer of this loss function is $F_k \in \mathbb{R}^{N \times k}$ such that $F_k F_k^\top$ contains the top-$k$ components of $\tilde{A}$'s eigen decomposition.

**A surrogate objective.** In practice, directly solving objective (5) can be computationally expensive for an extremely large matrix. To circumvent this, the feature representations can be equivalently recovered by minimizing the following contrastive learning objective [11], which can be efficiently trained end-to-end using a neural network:

$$\mathcal{L}(f) \triangleq -2\eta_u \mathcal{L}_1(f) - 2\eta_l \mathcal{L}_2(f) + \eta_u^2 \mathcal{L}_3(f) + 2\eta_u \eta_l \mathcal{L}_4(f) + \eta_l^2 \mathcal{L}_5(f), \tag{6}$$

where

$$\mathcal{L}_1(f) = \sum_{i \in \mathcal{Y}_l} \mathop{\mathbb{E}}_{\substack{\bar{x}_l \sim \mathbb{P}_{l_i}, \bar{x}_l' \sim \mathbb{P}_{l_i}, \\ x \sim \mathcal{T}(\cdot|\bar{x}_l), x^+ \sim \mathcal{T}(\cdot|\bar{x}_l')}} \left[ f(x)^\top f\left(x^+\right) \right], \mathcal{L}_2(f) = \mathop{\mathbb{E}}_{\substack{\bar{x}_u \sim \mathbb{P}, \\ x \sim \mathcal{T}(\cdot|\bar{x}_u), x^+ \sim \mathcal{T}(\cdot|\bar{x}_u)}} \left[ f(x)^\top f\left(x^+\right) \right],$$

$$\mathcal{L}_3(f) = \sum_{i,j \in \mathcal{Y}_l} \mathop{\mathbb{E}}_{\substack{\bar{x}_l \sim \mathbb{P}_{l_i}, \bar{x}_l' \sim \mathbb{P}_{l_j}, \\ x \sim \mathcal{T}(\cdot|\bar{x}_l), x^- \sim \mathcal{T}(\cdot|\bar{x}_l')}} \left[ \left( f(x)^\top f\left(x^-\right) \right)^2 \right],$$

$$\mathcal{L}_4(f) = \sum_{i \in \mathcal{Y}_l} \mathop{\mathbb{E}}_{\substack{\bar{x}_l \sim \mathbb{P}_{l_i}, \bar{x}_u \sim \mathbb{P}, \\ x \sim \mathcal{T}(\cdot|\bar{x}_l), x^- \sim \mathcal{T}(\cdot|\bar{x}_u)}} \left[ \left( f(x)^\top f\left(x^-\right) \right)^2 \right], \mathcal{L}_5(f) = \mathop{\mathbb{E}}_{\substack{\bar{x}_u \sim \mathbb{P}, \bar{x}_u' \sim \mathbb{P}, \\ x \sim \mathcal{T}(\cdot|\bar{x}_u), x^- \sim \mathcal{T}(\cdot|\bar{x}_u')}} \left[ \left( f(x)^\top f\left(x^-\right) \right)^2 \right].$$

Importantly, this contrastive loss allows drawing a theoretical equivalence between learned representations and the top-$k$ singular vectors of $\tilde{A}$, and facilitates theoretical understanding of the OOD generalization and detection on the data represented by $\tilde{A}$. The equivalence is formalized below.

**Theorem 3.3** (Theoretical equivalence between two objectives). *We define each row $f_x^\top$ of $F$ as a scaled version of learned feature embedding $f : \mathcal{X} \mapsto \mathbb{R}^k$, with $f_x = \sqrt{w_x} f(x)$. Then minimizing the loss function $\mathcal{L}_{mf}(F, A)$ in Equation 5 is equivalent to minimizing the surrogate loss in Equation 6. Full proof is in Appendix A.*

**Interpretation for OOD generalization and detection.** The loss learns feature representation jointly from both labeled ID data and unlabeled wild data, so that meaningful structures emerge for both OOD generalization and detection (e.g., covariate-shifted OOD data is embedded closely to the ID data, whereas semantic-shifted OOD data is distinguishable from ID data). At a high level, the loss components $\mathcal{L}_1$ and $\mathcal{L}_2$ contribute to pulling the embeddings of positive pairs closer, while $\mathcal{L}_3$, $\mathcal{L}_4$ and $\mathcal{L}_5$ push apart the embeddings of negative pairs. In particular, loss components on the positive pairs can pull together samples sharing the same classes, thereby helping OOD generalization. At

the same time, loss components on the negative pairs can help separate semantic OOD data in the embedding space, thus benefiting OOD detection.

**Difference from prior works**. Spectral contrastive learning has been employed to analyze problems such as self-supervised learning [7], unsupervised domain adaptation [12], novel category discovery [13], open-world semi-supervised learning [11] etc. These works share the underlying loss form by pulling together positive pairs and pushing away negative pairs. Despite the shared loss formulation, our work has fundamentally distinct data setup and learning goals, which focus on the joint OOD generalization and detection problems (*cf.* Section 2). We are interested in leveraging labeled ID data to classify both unlabeled ID and covariate OOD data correctly into the known categories while rejecting the remainder of unlabeled data from new categories, which was not studied in the prior works. Accordingly, we derive a novel theoretical analysis for our setup and present empirical verification uniquely tailored to our problem focus, which we present next.

# 4 Theoretical Analysis

In this section, we present a novel theoretical analysis of how the learned representations via graph spectral can facilitate both OOD generalization and detection.

## 4.1 Analytic Form of Learned Representations

To obtain the representations, one can train the neural network $f : \mathcal{X} \mapsto \mathbb{R}^k$ using the spectral loss defined in Equation 6. Minimizing the loss yields representation $Z \in \mathbb{R}^{N \times k}$, where each row vector $z_i = f(x_i)^\top$. According to Theorem 3.3, the closed-form solution for the representations is equivalent to performing spectral decomposition of the adjacency matrix. Thus, we have $F_k = \sqrt{D}Z$, where $F_k F_k^\top$ contains the top-$k$ components of $\tilde{A}$'s SVD decomposition and $D$ is the diagonal matrix. We further define the top-$k$ singular vectors of $\tilde{A}$ as $V_k \in \mathbb{R}^{N \times k}$, so we have $F_k = V_k \sqrt{\Sigma_k}$, where $\Sigma_k$ is a diagonal matrix of the top-$k$ singular values of $\tilde{A}$. By equalizing the two forms of $F_k$, the closed-formed solution of the learned feature space is given by $Z = [D]^{-\frac{1}{2}} V_k \sqrt{\Sigma_k}$.

## 4.2 Analysis Target

**Linear probing evaluation.** We assess OOD generalization performance based on the linear probing error, which is commonly used in self-supervised learning [6]. Specifically, the weight of a linear classifier is denoted as $\mathbf{M} \in \mathbb{R}^{k \times |\mathcal{Y}_l|}$, which is learned with ID data to minimize the error. The class prediction for an input $\bar{x}$ is given by $h(\bar{x}; f, \mathbf{M}) = \operatorname{argmax}_{i \in \mathcal{Y}_l}(f(\bar{x})^\top \mathbf{M})_i$. The linear probing error measures the misclassification of linear head on covariate-shifted OOD data:

$$\mathcal{E}(f) \triangleq \mathbb{E}_{\bar{x} \sim \mathbb{P}_{\text{out}}^{\text{covariate}}} \mathbb{1}[y(\bar{x}) \neq h(\bar{x}; f, \mathbf{M})], \tag{7}$$

where $y(\bar{x})$ indicates the ground-truth class of $\bar{x}$. $\mathcal{E}(f) = 0$ indicates perfect OOD generalization.

**Separability evaluation.** Based on the closed-form embeddings, we can also quantify the distance between the ID and semantic OOD data:

$$\mathcal{S}(f) \triangleq \mathbb{E}_{\bar{x}_i \sim \mathbb{P}_{\text{in}}, \bar{x}_j \sim \mathbb{P}_{\text{out}}^{\text{semantic}}} \|f(\bar{x}_i) - f(\bar{x}_j)\|_2^2. \tag{8}$$

The magnitude of $\mathcal{S}(f)$ reflects the extent of separation between ID and semantic OOD data. Larger $\mathcal{S}(f)$ suggests better OOD detection capability.

## 4.3 An Illustrative Example

**Setup.** We use an illustrative example to explain our theoretical insights. In Figure 2, the training examples come from 5 types of data: angel in sketch (ID), tiger in sketch (ID), angel in painting (covariate OOD), tiger in painting (covariate OOD), and panda (semantic OOD). The label space $\mathcal{Y}_l$ consists of two known classes: angel and tiger. Class Panda is considered a novel class. The goal is to classify between images of angels and tigers while rejecting images of pandas.

**Augmentation transformation probability.** Based on the data setup, we formally define the augmentation transformation, which encodes the probability of augmenting an original image $\bar{x}$

to the augmented view $x$:

$$\mathcal{T}(x \mid \bar{x}) = \begin{cases} \rho & \text{if} \quad y(\bar{x}) = y(x), d(\bar{x}) = d(x); \\ \alpha & \text{if} \quad y(\bar{x}) = y(x), d(\bar{x}) \neq d(x); \\ \beta & \text{if} \quad y(\bar{x}) \neq y(x), d(\bar{x}) = d(x); \\ \gamma & \text{if} \quad y(\bar{x}) \neq y(x), d(\bar{x}) \neq d(x). \end{cases} \quad (9)$$

Here $d(\bar{x})$ is the domain of sample $\bar{x}$, and $y(\bar{x})$ is the class label of sample $\bar{x}$. $\alpha$ indicates the augmentation probability when two samples share the same label but different domains, and $\beta$ indicates the probability when two samples share different class labels but with the same domain. It is natural to assume the magnitude order that follows $\rho \gg \max(\alpha, \beta) \geq \min(\alpha, \beta) \gg \gamma \geq 0$.

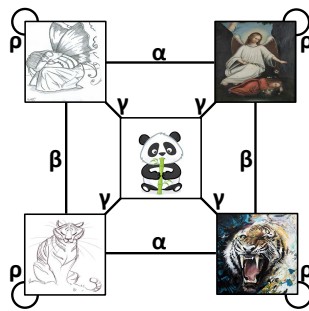

Figure 2: Illustration of graph and augmentation probability.

**Adjacency matrix.** With Eq. 9 and the definition in Section 3.1, we can derive the analytic form of adjacency matrix $A$.

$$\eta_u A^{(u)} = \begin{bmatrix} \rho^2 + \beta^2 + \alpha^2 + 2\gamma^2 & 2\rho\beta + \gamma^2 + 2\gamma\alpha & 2\rho\alpha + \gamma^2 + 2\gamma\beta & 2\alpha\beta + \gamma^2 + 2\gamma\rho & \gamma(\gamma + \alpha + \beta + 2\rho) \\ 2\rho\beta + \gamma^2 + 2\gamma\alpha & \rho^2 + \beta^2 + \alpha^2 + 2\gamma^2 & 2\alpha\beta + \gamma^2 + 2\gamma\rho & 2\rho\alpha + \gamma^2 + 2\gamma\beta & \gamma(\gamma + \alpha + \beta + 2\rho) \\ 2\rho\alpha + \gamma^2 + 2\gamma\beta & 2\alpha\beta + \gamma^2 + 2\gamma\rho & \rho^2 + \beta^2 + \alpha^2 + 2\gamma^2 & 2\rho\beta + \gamma^2 + 2\gamma\alpha & \gamma(\gamma + \alpha + \beta + 2\rho) \\ 2\alpha\beta + \gamma^2 + 2\gamma\rho & 2\rho\alpha + \gamma^2 + 2\gamma\beta & 2\rho\beta + \gamma^2 + 2\gamma\alpha & \rho^2 + \beta^2 + \alpha^2 + 2\gamma^2 & \gamma(\gamma + \alpha + \beta + 2\rho) \\ \gamma(\gamma + \alpha + \beta + 2\rho) & \gamma(\gamma + \alpha + \beta + 2\rho) & \gamma(\gamma + \alpha + \beta + 2\rho) & \gamma(\gamma + \alpha + \beta + 2\rho) & \rho^2 + 4\gamma^2 \end{bmatrix}$$

$$(10)$$

$$A = \frac{1}{C}(\eta_l A^{(l)} + \eta_u A^{(u)}) = \frac{1}{C}\left( \begin{bmatrix} \rho^2 + \beta^2 & 2\rho\beta & \rho\alpha + \gamma\beta & \alpha\beta + \gamma\rho & \gamma(\rho + \beta) \\ 2\rho\beta & \rho^2 + \beta^2 & \alpha\beta + \gamma\rho & \rho\alpha + \gamma\beta & \gamma(\rho + \beta) \\ \rho\alpha + \gamma\beta & \alpha\beta + \gamma\rho & \alpha^2 + \gamma^2 & 2\gamma\alpha & \gamma(\alpha + \gamma) \\ \alpha\beta + \gamma\rho & \rho\alpha + \gamma\beta & 2\gamma\alpha & \alpha^2 + \gamma^2 & \gamma(\alpha + \gamma) \\ \gamma(\rho + \beta) & \gamma(\rho + \beta) & \gamma(\alpha + \gamma) & \gamma(\alpha + \gamma) & 2\gamma^2 \end{bmatrix} + \eta_u A^{(u)} \right),$$

$$(11)$$

where $C$ is the normalization constant to ensure the summation of weights amounts to 1. Each row or column encodes connectivity associated with a specific sample, ordered by: angel sketch, tiger sketch, angel painting, tiger painting, and panda. We refer readers to Appendix D.1 for the detailed derivation.

**Main analysis.** We are primarily interested in analyzing the representation space derived from $A$. We mainly put analysis on the top-3 eigenvectors $\widehat{V} \in \mathbb{R}^{5 \times 3}$ and measure both the linear probing error and separability. The full derivation of Theorem 4.1 and Theorem 4.2 can be found in Appendix D.1.

**Theorem 4.1.** *Assume* $\eta_u = 5, \eta_l = 1$, *we have:*

$$\widehat{V} = \begin{cases} \begin{bmatrix} \frac{1}{\sqrt{3}} & \frac{1}{\sqrt{3}} & \frac{1}{\sqrt{6}} & \frac{1}{\sqrt{6}} & 0 \\ 0 & 0 & 0 & 0 & 1 \\ -\frac{1}{\sqrt{3}} & \frac{1}{\sqrt{3}} & -\frac{1}{\sqrt{6}} & \frac{1}{\sqrt{6}} & 0 \end{bmatrix}^{\top} & \text{, if } \frac{9}{8}\alpha > \beta; \\[2em] \begin{bmatrix} \frac{1}{\sqrt{3}} & \frac{1}{\sqrt{3}} & \frac{1}{\sqrt{6}} & \frac{1}{\sqrt{6}} & 0 \\ 0 & 0 & 0 & 0 & 1 \\ -\frac{1}{\sqrt{6}} & -\frac{1}{\sqrt{6}} & \frac{1}{\sqrt{3}} & \frac{1}{\sqrt{3}} & 0 \end{bmatrix}^{\top} & \text{, if } \frac{9}{8}\alpha < \beta. \end{cases}, \quad \mathcal{E}(f) = \begin{cases} 0 & \text{, if } \frac{9}{8}\alpha > \beta; \\[1em] 2 & \text{, if } \frac{9}{8}\alpha < \beta. \end{cases} \quad (12)$$

**Interpretation.** The discussion can be divided into two cases: (1) $\frac{9}{8}\alpha > \beta$. (2) $\frac{9}{8}\alpha < \beta$. In the first case when the connection between the class (multiplied by $\frac{9}{8}$) is stronger than the domain, the model could learn a perfect ID classifier based on features in the first two rows in $V$ and effectively generalize to the covariate-shifted domain (the third and fourth row in $\widehat{V}$), achieving perfect OOD generalization with linear probing error $\mathcal{E}(f) = 0$. In the second case when the connection between the domain is stronger than the connection between the class (scaled by $\frac{9}{8}$), the embeddings of covariate-shifted OOD data are identical, resulting in high OOD generalization error.

**Theorem 4.2.** *Denote* $\alpha' = \frac{\alpha}{\rho}$ *and* $\beta' = \frac{\beta}{\rho}$ *and assume* $\eta_u = 5, \eta_l = 1$, *we have:*

$$\mathcal{S}(f) = \begin{cases} (7 + 12\beta' + 12\alpha')(\frac{1-2\beta'}{3}(1 - \beta' - \frac{3}{4}\alpha')^2 + 1) & \text{, if } \frac{9}{8}\alpha > \beta; \\ (7 + 12\beta' + 12\alpha')(\frac{2-3\alpha'}{8}(1 - \beta' - \frac{3}{4}\alpha')^2 + 1) & \text{, if } \frac{9}{8}\alpha < \beta. \end{cases} \quad (13)$$

**Interpretation.** We analyze the function $S(f)$ under different $\alpha'$ and $\beta'$ values in Figure 3. Overall the distance between semantic OOD data and ID data displays a large value, which facilitates OOD detection. Note that a clear boundary in Figure 3 indicates $\frac{9}{8}\alpha = \beta$.

**More analysis.** Building upon the understanding of both OOD generalization and detection, we further discuss the influence of different semantic OOD data in Appendix B, and the impact of ID labels in Appendix C.

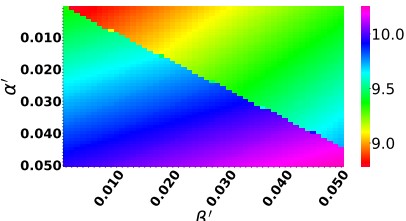

Figure 3: Value of function $S(f)$

## 5 Experiments

Beyond theoretical insights, we show empirically that our approach is competitive. We present the experimental setup in Section 5.1, results in Section 5.2, and further analysis in Section 5.3.

### 5.1 Experimental Setup

**Datasets and benchmarks.** Following the setup of [5], we employ CIFAR-10 [14] as $\mathbb{P}_{\text{in}}$ and CIFAR-10-C [15] with Gaussian additive noise as the $\mathbb{P}_{\text{out}}^{\text{covariate}}$. For $\mathbb{P}_{\text{out}}^{\text{semantic}}$, we leverage SVHN [16], LSUN [17], Places365 [18], Textures [19]. To simulate the wild distribution $\mathbb{P}_{\text{wild}}$, we adopt the same mixture ratio as in Scone [5], where $\pi_c = 0.5$ and $\pi_s = 0.1$. Detailed descriptions of the datasets and data mixture can be found in the Appendix E.1. To demonstrate the adaptability and robustness of our proposed method, we extend the framework to more diverse and challenging datasets. Large-scale results on the ImageNet dataset can be found in Appendix E.2. Additional results on the Office-Home [20] can be found in Appendix E.3. More ablation studies can be found in Appendix E.4.

**Implementation details.** We adopt Wide ResNet with 40 layers and a widen factor of 2 [21]. We use stochastic gradient descent with Nesterov momentum [22], with weight decay 0.0005 and momentum 0.09. We divide CIFAR-10 training set into 50% labeled as ID and 50% unlabeled. And we mix unlabeled CIFAR-10, CIFAR-10-C, and semantic OOD data to generate the wild dataset. Starting from random initialization, we train the network with the loss function in Eq. 6 for 1000 epochs. The learning rate is 0.03 and the batch size is 512. $\eta_u$ is selected within $\{1.00, 2.00\}$ and $\eta_l$ is within $\{0.02, 0.10, 0.50, 1.00\}$. Subsequently, we follow the standard approach [12] and use labeled ID data to fine-tune the model with cross-entropy loss for better generalization ability. We fine-tune for 20 epochs with a learning rate of 0.005 and batch size of 512. The fine-tuned model is used to evaluate the OOD generalization and OOD detection performance. We utilize a distance-based method for OOD detection, which resonates with our theoretical analysis. Specifically, our default approach employs a simple non-parametric KNN distance [23], which does not impose any distributional assumption on the feature space. The threshold is determined based on the clean ID set at 95% percentile. For further implementation details, hyper-parameters, and validation strategy, please see Appendix F.

### 5.2 Results and Discussion

**Competitive empirical performance.** The main results in Table 1 demonstrate that our method not only enjoys theoretical guarantees but also exhibits competitive empirical performance compared to existing baselines. For a comprehensive evaluation, we consider three groups of methods for OOD generalization and OOD detection. Closest to our setting, we compare with strong baselines trained with wild data, namely OE [36], Energy-regularized learning [26], Woods [37], and Scone [5].

The empirical results provide interesting insights into the performance of various methods for OOD detection and generalization. **(1)** Methods tailored for OOD detection tend to capture the domain-variant information and struggle with the covariate distribution shift, resulting in suboptimal OOD accuracy. **(2)** While approaches for OOD generalization demonstrate improved OOD accuracy, they cannot effectively distinguish between ID data and semantic OOD data, leading to poor OOD detection performance. **(3)** Methods trained with wild data emerge as robust OOD detectors, yet display a notable decline in OOD generalization, highlighting the confusion introduced by covariate

| Method | SVHN $\mathbb{P}_{out}^{semantic}$, CIFAR-10-C $\mathbb{P}_{out}^{covariate}$ | | | | LSUN-C $\mathbb{P}_{out}^{semantic}$, CIFAR-10-C $\mathbb{P}_{out}^{covariate}$ | | | | Textures $\mathbb{P}_{out}^{semantic}$, CIFAR-10-C $\mathbb{P}_{out}^{covariate}$ | | | |
|---|---|---|---|---|---|---|---|---|---|---|---|---|
| | OOD Acc.↑ | ID Acc.↑ | FPR↓ | AUROC↑ | OOD Acc.↑ | ID Acc.↑ | FPR↓ | AUROC↑ | OOD Acc.↑ | ID Acc.↑ | FPR↓ | AUROC↑ |
| *OOD detection* | | | | | | | | | | | | |
| **MSP** [24] | 75.05 | 94.84 | 48.49 | 91.89 | 75.05 | 94.84 | 30.80 | 95.65 | 75.05 | 94.84 | 59.28 | 88.50 |
| **ODIN** [25] | 75.05 | 94.84 | 33.35 | 91.96 | 75.05 | 94.84 | 15.52 | 97.04 | 75.05 | 94.84 | 49.12 | 84.97 |
| **Energy** [26] | 75.05 | 94.84 | 35.59 | 90.96 | 75.05 | 94.84 | 8.26 | 98.35 | 75.05 | 94.84 | 52.79 | 85.22 |
| **Mahalanobis** [27] | 75.05 | 94.84 | 12.89 | 97.62 | 75.05 | 94.84 | 39.22 | 94.15 | 75.05 | 94.84 | 15.00 | 97.33 |
| **ViM** [28] | 75.05 | 94.84 | 21.95 | 95.48 | 75.05 | 94.84 | 5.90 | 98.82 | 75.05 | 94.84 | 29.35 | 93.70 |
| **KNN** [23] | 75.05 | 94.84 | 28.92 | 95.71 | 75.05 | 94.84 | 28.08 | 95.33 | 75.05 | 94.84 | 39.50 | 92.73 |
| **ASH** [29] | 75.05 | 94.84 | 40.76 | 90.16 | 75.05 | 94.84 | 2.39 | 99.35 | 75.05 | 94.84 | 53.37 | 85.63 |
| *OOD generalization* | | | | | | | | | | | | |
| **ERM** [30] | 75.05 | 94.84 | 35.59 | 90.96 | 75.05 | 94.84 | 8.26 | 98.35 | 75.05 | 94.84 | 52.79 | 85.22 |
| **IRM** [31] | 77.92 | 90.85 | 63.65 | 90.70 | 77.92 | 90.85 | 36.67 | 94.20 | 77.92 | 90.85 | 59.42 | 87.81 |
| **Mixup** [32] | 79.17 | 93.30 | 97.33 | 18.78 | 79.17 | 93.30 | 52.10 | 76.66 | 79.17 | 93.30 | 58.24 | 75.70 |
| **VREx** [33] | 76.90 | 91.35 | 55.92 | 91.22 | 76.90 | 91.35 | 51.50 | 91.56 | 76.90 | 91.35 | 65.45 | 85.46 |
| **EQRM** [34] | 75.71 | 92.93 | 51.86 | 90.92 | 75.71 | 92.93 | 21.53 | 96.49 | 75.71 | 92.93 | 57.18 | 89.11 |
| **SharpDRO** [35] | 79.03 | 94.91 | 21.24 | 96.14 | 79.03 | 94.91 | 5.67 | 98.71 | 79.03 | 94.91 | 42.94 | 89.99 |
| *Learning w. $\mathbb{P}_{wild}$* | | | | | | | | | | | | |
| **OE** [36] | 37.61 | 94.68 | 0.84 | 99.80 | 41.37 | 93.99 | 3.07 | 99.26 | 44.71 | 92.84 | 29.36 | 93.93 |
| **Energy (w. outlier)** [26] | 20.74 | 90.22 | 0.86 | 99.81 | 32.55 | 92.97 | 2.33 | 99.93 | 49.34 | 94.68 | 16.42 | 96.46 |
| **Woods** [37] | 52.76 | 94.86 | 2.11 | 99.52 | 76.90 | 95.02 | 1.80 | 99.56 | 83.14 | 94.49 | 39.10 | 90.45 |
| **Scone** [5] | 84.69 | 94.65 | 10.86 | 97.84 | 84.58 | 93.73 | 10.23 | 98.02 | **85.56** | 93.97 | 37.15 | 90.91 |
| **Ours** | **86.62**$_{\pm0.3}$ | 93.10$_{\pm0.1}$ | **0.13**$_{\pm0.0}$ | **99.98**$_{\pm0.0}$ | **85.88**$_{\pm0.2}$ | 92.61$_{\pm0.1}$ | **1.76**$_{\pm0.8}$ | 99.75$_{\pm0.1}$ | 81.40$_{\pm0.7}$ | 92.50$_{\pm0.1}$ | **12.05**$_{\pm0.8}$ | **98.25**$_{\pm0.2}$ |

Table 1: Main results: comparison with competitive OOD generalization and OOD detection methods on CIFAR-10. Additional results for the Places365 and LSUN-R datasets can be found in Table 3. **Bold**=best. (*Since all the OOD detection methods use the same model trained with the CE loss on $\mathbb{P}_{in}$, they display the same ID and OOD accuracy on CIFAR-10-C.)

OOD data. In contrast, our method excels in both OOD detection and generalization performance. Our method even surpasses the latest method Scone by **25.10**% in terms of FPR95 on the Textures dataset. Methodologically, Scone uses constrained optimization whereas our method brings a novel graph-theoretic perspective. More results can be found in the Appendix E.

**Better adaptation to the heterogeneous distribution.** We conduct a comparative analysis of our methods against other state-of-the-art spectral learning approaches within their respective domains. Specifically, Haochen et al. [7] investigate unsupervised learning, Shen et al. [12] delve into unsupervised domain adaptation, and Sun et al. [13] explores novel class discovery. The baseline methods all assume unlabeled data exhibits a homogeneous distribution, either entirely from $\mathbb{P}_{out}^{covariate}$ in the case of unsupervised domain adaptation or entirely from $\mathbb{P}_{out}^{semantic}$ in the case

| $\mathbb{P}_{out}^{semantic}$ | Method | OOD Acc.↑ | ID Acc.↑ | FPR↓ | AUROC↑ |
|---|---|---|---|---|---|
| | SCL [7, 12] | 75.96 | 87.58 | 21.53 | 96.56 |
| SVHN | NSCL [13] | 85.49 | 92.42 | 0.15 | 99.97 |
| | Ours | **86.62** | **93.10** | **0.13** | **99.98** |
| | SCL [7, 12] | 65.48 | 85.14 | 81.30 | 83.34 |
| LSUN-C | NSCL [13] | 77.64 | 90.61 | 18.43 | 97.84 |
| | Ours | **85.88** | **92.61** | **1.76** | **99.75** |
| | SCL [7, 12] | 63.05 | 83.07 | 66.86 | 87.59 |
| TEXTURES | NSCL [13] | 62.86 | 86.56 | 39.04 | 92.59 |
| | Ours | **81.40** | **92.50** | **12.05** | **98.25** |

Table 2: Comparison with spectral learning methods.

of novel class discovery. As depicted in Table 2, our results reveal a significant improvement over competing baselines on both OOD generalization and detection. We attribute this empirical success to our better adaptation to the heterogeneous mixture of wild distributions. Additional results can be found in Table 6. More ablation studies can be found in Appendix E.4.

## 5.3 Further Analysis

**Visualization of OOD detection score distributions.** In Figure 4 (a), we visualize the distribution of KNN distances. The KNN scores are computed based on samples from the test set after contrastive training and fine-tuning stages. There are two salient observations: First, our learning framework effectively pushes the semantic OOD data to be apart from the ID data in the embedding space, which benefits OOD detection. Moreover, as evidenced by the small KNN distance, covariate-shifted OOD data is embedded closely to the ID data, which aligns with our expectations.

**Visualization of embeddings.** Figure 4 (b) displays the t-SNE [38] visualization of the normalized penultimate-layer embeddings. Samples are from the test set of ID, covariate OOD, and semantic OOD data, respectively. The visualization demonstrates the alignment of ID and covariate OOD data in the embedding space, which allows the classifier learned on the ID data to extrapolate to the covariate OOD data thereby benefiting OOD generalization.

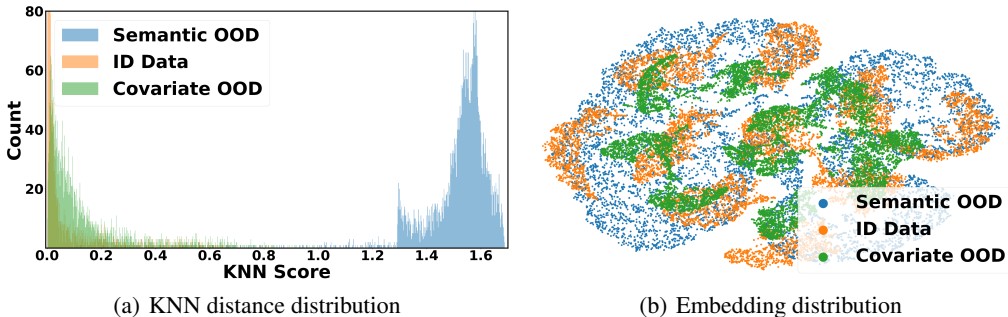

|                                    |                                    |
|:----------------------------------:|:----------------------------------:|
| (a) KNN distance distribution      | (b) Embedding distribution         |

Figure 4: (a) Distribution of KNN distance. (b) t-SNE visualization of learned embeddings. We employ CIFAR-10 as $\mathbb{P}_{\text{in}}$, CIFAR-10-C as $\mathbb{P}_{\text{out}}^{\text{covariate}}$, and SVHN as $\mathbb{P}_{\text{out}}^{\text{semantic}}$.

## 6 Related Works

**Out-of-distribution detection.** OOD detection has gained soaring research attention in recent years. The current research track can be divided into post hoc and regularization-based methods. Post hoc methods derive OOD scores at test-time based on a pre-trained model, which can be categorized as confidence-based methods [39, 24, 40], energy-based methods [26, 41, 42, 43, 44, 29], distance-based methods [45, 46, 47, 23, 48, 49, 50], and gradient-based method [51]. On the other hand, regularization-based methods aim to train the OOD detector by training-time regularization. Most approaches require auxiliary OOD data [52, 53, 54, 36, 55, 56, 57]. However, a limitation of existing methods is the reliance on clean semantic OOD datasets for training. To address this challenge, WOODS [37] first explored the use of wild data, which includes unlabeled ID and semantic OOD data. Building upon this idea, SCONE [5] extended the characterization of wild data to encompass ID, covariate OOD, and semantic OOD data, providing a more generalized data mixture in practice. In our paper, we provide a novel graph-theoretic approach for understanding both OOD generalization and detection based on the setup proposed by Scone [5].

**Out-of-distribution generalization.** OOD generalization aims to learn domain-invariant representations that can effectively generalize to unseen domains, which is more challenging than classic domain adaptation problem [58, 59, 60, 61], where the model has access to unlabeled data from the target domain. OOD generalization and domain generalization [62] focus on capturing semantic features that remain consistent across diverse domains, which can be categorized as reducing feature discrepancies across the source domains [63, 64, 31, 65, 66, 67], ensemble and meta learning [68, 69, 70, 71, 72], robust optimization [73, 74, 75, 76, 77], augmentation [78, 79, 80, 81], and disentanglement [82]. Distinct from prior literature about generalization, Scone [5] introduces a framework that leverages the wild data ubiquitous in the real world, aiming to build a robust classifier and a reliable OOD detector simultaneously. Following the same problem setting in [5], we contribute novel theoretical insights into the understanding of both OOD generalization and detection.

**Spectral graph theory.** Spectral graph theory is a classical research field [8, 83, 84, 85, 86], concerning the study of graph partitioning through analyzing the eigenspace of the adjacency matrix. The spectral graph theory is also widely applied in machine learning [87, 88, 89, 90, 91, 92]. Recently, Haochen et al. [7] presented unsupervised spectral contrastive loss derived from the factorization of the graph's adjacency matrix. Shen et al. [12] provided a graph-theoretic analysis for unsupervised domain adaptation based on the assumption of unlabeled data entirely from $\mathbb{P}_{\text{out}}^{\text{covariate}}$. Sun et al. [13] first introduced the label information and explored novel category discovery, considering unlabeled data covers $\mathbb{P}_{\text{out}}^{\text{semantic}}$. All of the previous literature assumed unlabeled data has a homogeneous distribution. In contrast, our work focuses on the joint problem of OOD generalization and detection, tackling the challenge of unlabeled data characterized by a heterogeneous mixture distribution, which is a more general and complex scenario than previous works.

**Contrastive learning.** Recent works on contrastive learning advance the development of deep neural networks with a huge empirical success [6, 93, 94, 95, 96, 97, 98, 99, 100, 101]. Simultaneously, many theoretical works establish the foundation for understanding representations learned by contrastive learning through linear probing evaluation [102, 103, 104, 105, 106, 107]. Haochen et

al. [7, 108], Sun et al. [13] extended the understanding and providing error analyses for different downstream tasks. Orthogonal to prior works, we provide a graph-theoretic framework tailored for the wild environment to understand both OOD generalization and detection.

## 7    Conclusion

In this paper, we present a graph-theoretic framework to jointly tackle both OOD generalization and detection problems. Based on the graph formulation, the data representations can be derived by factorizing the graph's adjacency matrix, allowing us to draw theoretical insight into both OOD generalization and detection performance. In particular, we analyze the closed-form solutions of linear probing error for OOD generalization, as well as separability quantifying OOD detection capability via the distance between the ID and semantic OOD data. Empirically, our framework demonstrates competitive performance against existing baselines, closely aligning with our theoretical insights. We anticipate that our theoretical framework and findings will inspire further research in unifying and understanding both OOD generalization and detection.

## 8    Broader Impact

In the rapidly evolving landscape of machine learning, addressing the dual challenges of OOD generalization and detection has become paramount for deploying *robust and reliable* models in real-world scenarios. Our work provides a novel spectral learning solution, which not only improves model performance but also ensures its reliability and safety in diverse, dynamic environments. The implications of our research extend beyond theoretical advancements, with potential applications in healthcare, autonomous systems, and finance. The ability to deploy models with superior OOD generalization and detection capabilities addresses a critical bottleneck in the adoption of machine learning technologies, fostering trust among end-users and stakeholders.

## 9    Limitations

In our experimental setup, we focus on covariate shift as the primary form of shift in the out-of-distribution (OOD) generalization problem, a topic extensively explored in the literature. However, it's important to acknowledge the existence of other types of distributional shifts (e.g., concept shift), which we defer for future investigation.

### Acknowledgement

We thank Yiyou Sun for the valuable discussion and input during the project. Li gratefully acknowledges the funding support by the AFOSR Young Investigator Program under award number FA9550-23-1-0184, National Science Foundation (NSF) Award No. IIS-2237037 & IIS-2331669, Office of Naval Research under grant number N00014-23-1-2643, Philanthropic Fund from SFF, and faculty research awards/gifts from Google and Meta.

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

# A   Technical Details of Spectral Learning

*Proof.* We can expand $\mathcal{L}_{\mathrm{mf}}(F, A)$ and obtain

$$
\begin{aligned}
\mathcal{L}_{\mathrm{mf}}(F, A) &= \sum_{x,x'\in\mathcal{X}} \left( \frac{w_{xx'}}{\sqrt{w_x w_{x'}}} - f_x^\top f_{x'} \right)^2 \\
&= \mathrm{const} + \sum_{x,x'\in\mathcal{X}} \left( -2 w_{xx'} f(x)^\top f(x') + w_x w_{x'} \left( f(x)^\top f(x') \right)^2 \right),
\end{aligned}
$$

where $f_x = \sqrt{w_x} f(x)$ is a re-scaled version of $f(x)$. At a high level, we follow the proof in Haochen et al. [7], while the specific form of loss varies with the different definitions of positive/negative pairs. The form of $\mathcal{L}(f)$ is derived from plugging $w_{xx'}$ and $w_x$.

Recall that $w_{xx'}$ is defined by

$$
w_{xx'} = \eta_l \sum_{i\in\mathcal{Y}_l} \mathbb{E}_{\bar{x}_l\sim\mathbb{P}_{l_i}} \mathbb{E}_{\bar{x}_l'\sim\mathbb{P}_{l_i}} \mathcal{T}(x|\bar{x}_l)\mathcal{T}(x'|\bar{x}_l') + \eta_u \mathbb{E}_{\bar{x}_u\sim\mathbb{P}} \mathcal{T}(x|\bar{x}_u)\mathcal{T}(x'|\bar{x}_u),
$$

and $w_x$ is given by

$$
\begin{aligned}
w_x &= \sum_{x'} w_{xx'} \\
&= \eta_l \sum_{i\in\mathcal{Y}_l} \mathbb{E}_{\bar{x}_l\sim\mathbb{P}_{l_i}} \mathbb{E}_{\bar{x}_l'\sim\mathbb{P}_{l_i}} \mathcal{T}(x|\bar{x}_l) \sum_{x'} \mathcal{T}(x'|\bar{x}_l') + \eta_u \mathbb{E}_{\bar{x}_u\sim\mathbb{P}} \mathcal{T}(x|\bar{x}_u) \sum_{x'} \mathcal{T}(x'|\bar{x}_u) \\
&= \eta_l \sum_{i\in\mathcal{Y}_l} \mathbb{E}_{\bar{x}_l\sim\mathbb{P}_{l_i}} \mathcal{T}(x|\bar{x}_l) + \eta_u \mathbb{E}_{\bar{x}_u\sim\mathbb{P}} \mathcal{T}(x|\bar{x}_u).
\end{aligned}
$$

Plugging in $w_{xx'}$ we have,

$$
\begin{aligned}
&- 2 \sum_{x,x'\in\mathcal{X}} w_{xx'} f(x)^\top f(x') \\
={}&- 2 \sum_{x,x^+\in\mathcal{X}} w_{xx^+} f(x)^\top f(x^+) \\
={}&- 2\eta_l \sum_{i\in\mathcal{Y}_l} \mathbb{E}_{\bar{x}_l\sim\mathbb{P}_{l_i}} \mathbb{E}_{\bar{x}_l'\sim\mathbb{P}_{l_i}} \sum_{x,x'\in\mathcal{X}} \mathcal{T}(x|\bar{x}_l)\mathcal{T}(x'|\bar{x}_l') f(x)^\top f(x') \\
&- 2\eta_u \mathbb{E}_{\bar{x}_u\sim\mathbb{P}} \sum_{x,x'} \mathcal{T}(x|\bar{x}_u)\mathcal{T}(x'|\bar{x}_u) f(x)^\top f(x') \\
={}&- 2\eta_l \sum_{\substack{i\in\mathcal{Y}_l}} \mathbb{E}_{\substack{\bar{x}_l\sim\mathbb{P}_{l_i},\bar{x}_l'\sim\mathbb{P}_{l_i},\\ x\sim\mathcal{T}(\cdot|\bar{x}_l),x^+\sim\mathcal{T}(\cdot|\bar{x}_l')}} \left[ f(x)^\top f(x^+) \right] \\
&- 2\eta_u \mathbb{E}_{\substack{\bar{x}_u\sim\mathbb{P},\\ x\sim\mathcal{T}(\cdot|\bar{x}_u),x^+\sim\mathcal{T}(\cdot|\bar{x}_u)}} \left[ f(x)^\top f(x^+) \right] \\
={}&- 2\eta_l \mathcal{L}_1(f) - 2\eta_u \mathcal{L}_2(f).
\end{aligned}
$$

Plugging $w_x$ and $w_{x'}$ we have,

$$\sum_{x,x'\in\mathcal{X}} w_x w_{x'} \left(f(x)^\top f(x')\right)^2$$

$$=\sum_{x,x^-\in\mathcal{X}} w_x w_{x^-} \left(f(x)^\top f(x^-)\right)^2$$

$$=\sum_{x,x'\in\mathcal{X}} \left(\eta_l \sum_{i\in\mathcal{Y}_l} \mathbb{E}_{\bar{x}_l\sim\mathbb{P}_{l_i}}\mathcal{T}(x|\bar{x}_l) + \eta_u \mathbb{E}_{\bar{x}_u\sim\mathbb{P}}\mathcal{T}(x|\bar{x}_u)\right)$$

$$\cdot \left(\eta_l \sum_{j\in\mathcal{Y}_l} \mathbb{E}_{\bar{x}'_l\sim\mathbb{P}_{l_j}}\mathcal{T}(x^-|\bar{x}'_l) + \eta_u \mathbb{E}_{\bar{x}'_u\sim\mathbb{P}}\mathcal{T}(x^-|\bar{x}'_u)\right) \left(f(x)^\top f(x^-)\right)^2$$

$$=\eta_l^2 \sum_{x,x^-\in\mathcal{X}} \sum_{i\in\mathcal{Y}_l} \mathbb{E}_{\bar{x}_l\sim\mathbb{P}_{l_i}}\mathcal{T}(x|\bar{x}_l) \sum_{j\in\mathcal{Y}_l} \mathbb{E}_{\bar{x}'_l\sim\mathbb{P}_{l_j}}\mathcal{T}(x^-|\bar{x}'_l) \left(f(x)^\top f(x^-)\right)^2$$

$$+ 2\eta_l\eta_u \sum_{x,x^-\in\mathcal{X}} \sum_{i\in\mathcal{Y}_l} \mathbb{E}_{\bar{x}_l\sim\mathbb{P}_{l_i}}\mathcal{T}(x|\bar{x}_l) \mathbb{E}_{\bar{x}_u\sim\mathbb{P}}\mathcal{T}(x^-|\bar{x}_u) \left(f(x)^\top f(x^-)\right)^2$$

$$+ \eta_u^2 \sum_{x,x^-\in\mathcal{X}} \mathbb{E}_{\bar{x}_u\sim\mathbb{P}}\mathcal{T}(x|\bar{x}_u) \mathbb{E}_{\bar{x}'_u\sim\mathbb{P}}\mathcal{T}(x^-|\bar{x}'_u) \left(f(x)^\top f(x^-)\right)^2$$

$$=\eta_l^2 \sum_{i\in\mathcal{Y}_l}\sum_{j\in\mathcal{Y}_l} \mathop{\mathbb{E}}_{\substack{\bar{x}_l\sim\mathbb{P}_{l_i},\bar{x}'_l\sim\mathbb{P}_{l_j}, \\ x\sim\mathcal{T}(\cdot|\bar{x}_l),x^-\sim\mathcal{T}(\cdot|\bar{x}'_l)}} \left[\left(f(x)^\top f(x^-)\right)^2\right]$$

$$+ 2\eta_l\eta_u \sum_{i\in\mathcal{Y}_l} \mathop{\mathbb{E}}_{\substack{\bar{x}_l\sim\mathbb{P}_{l_i},\bar{x}_u\sim\mathbb{P}, \\ x\sim\mathcal{T}(\cdot|\bar{x}_l),x^-\sim\mathcal{T}(\cdot|\bar{x}_u)}} \left[\left(f(x)^\top f(x^-)\right)^2\right]$$

$$+ \eta_u^2 \mathop{\mathbb{E}}_{\substack{\bar{x}_u\sim\mathbb{P},\bar{x}'_u\sim\mathbb{P}, \\ x\sim\mathcal{T}(\cdot|\bar{x}_u),x^-\sim\mathcal{T}(\cdot|\bar{x}'_u)}} \left[\left(f(x)^\top f(x^-)\right)^2\right]$$

$$=\eta_l^2 \mathcal{L}_3(f) + 2\eta_l\eta_u \mathcal{L}_4(f) + \eta_u^2 \mathcal{L}_5(f).$$

$\square$

# B    Impact of Semantic OOD Data

In our main analysis in Section 4, we consider semantic OOD to be from a different domain. Alternatively, instances of semantic OOD data can come from the same domain as covariate OOD data. In this section, we provide a complete picture by contrasting these two cases.

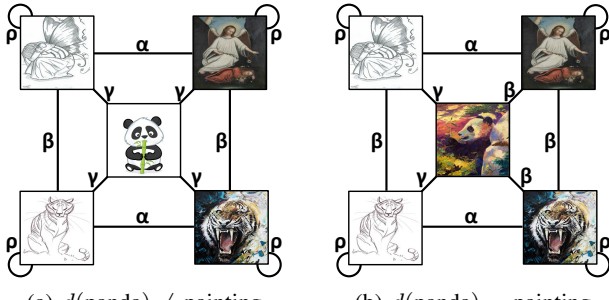

(a) $d(\text{panda}) \neq \text{painting}$     (b) $d(\text{panda}) = \text{painting}$

Figure 5: Illustration of 5 nodes graph and the augmentation probability defined by classes and domains. Figure (a) illustrates the scenario where semantic OOD data has a different domain from covariate OOD. Figure (b) depicts the case where semantic OOD and covariate OOD share the same domain.

**Setup.** In Figure 5, we illustrate two scenarios where the semantic OOD data has either a different or the same domain label as covariate OOD data. Other setups are the same as Sec. 4.3.

**Adjacency matrix.** The adjacency matrix for scenario (a) has been derived in Eq. 11. For the alternative scenario (b) where semantic OOD shares the same domain as the covariate OOD, we can derive the analytic form of adjacency matrix $A_1$.

$$\eta_u A_1^{(u)} = \begin{bmatrix} \rho^2+\beta^2+\alpha^2+2\gamma^2 & 2\rho\beta+\gamma^2+2\gamma\alpha & 2\rho\alpha+3\gamma\beta & 2\alpha\beta+\gamma\beta+2\gamma\rho & \alpha\beta+2\gamma(\beta+\rho) \\ 2\rho\beta+\gamma^2+2\gamma\alpha & \rho^2+\beta^2+\alpha^2+2\gamma^2 & 2\alpha\beta+\gamma\beta+2\gamma\rho & 2\rho\alpha+3\gamma\beta & \alpha\beta+2\gamma(\beta+\rho) \\ 2\rho\alpha+3\gamma\beta & 2\alpha\beta+\gamma\beta+2\gamma\rho & \rho^2+2\beta^2+\alpha^2+\gamma^2 & 2\rho\beta+\beta^2+2\gamma\alpha & 2\rho\beta+\beta^2+\gamma^2+\gamma\alpha \\ 2\alpha\beta+\gamma\beta+2\gamma\rho & 2\alpha\rho+3\gamma\beta & 2\rho\beta+\beta^2+2\gamma\alpha & \rho^2+2\beta^2+\alpha^2+\gamma^2 & 2\rho\beta+\beta^2+\gamma^2+\gamma\alpha \\ \alpha\beta+2\gamma(\beta+\rho) & \alpha\beta+2\gamma(\beta+\rho) & 2\rho\beta+\beta^2+\gamma^2+\gamma\alpha & 2\rho\beta+\beta^2+\gamma^2+\gamma\alpha & \rho^2+2\beta^2+2\gamma^2 \end{bmatrix} \tag{14}$$

$$A_1 = \frac{1}{C_1}(\eta_l A_1^{(l)} + \eta_u A_1^{(u)}) = \frac{1}{C_1}\left(\begin{bmatrix} \rho^2+\beta^2 & 2\rho\beta & \rho\alpha+\gamma\beta & \alpha\beta+\gamma\rho & \gamma(\beta+\rho) \\ 2\rho\beta & \rho^2+\beta^2 & \alpha\beta+\gamma\rho & \rho\alpha+\gamma\beta & \gamma(\beta+\rho) \\ \rho\alpha+\gamma\beta & \alpha\beta+\gamma\rho & \alpha^2+\gamma^2 & 2\gamma\alpha & \gamma(\gamma+\alpha) \\ \alpha\beta+\gamma\rho & \rho\alpha+\gamma\beta & 2\gamma\alpha & \alpha^2+\gamma^2 & \gamma(\gamma+\alpha) \\ \gamma(\beta+\rho) & \gamma(\beta+\rho) & \gamma(\gamma+\alpha) & \gamma(\gamma+\alpha) & 2\gamma^2 \end{bmatrix} + \eta_u A_1^{(u)}\right), \tag{15}$$

where $C_1$ is the normalization constant to ensure the summation of weights amounts to 1. Each row or column encodes connectivity associated with a specific sample, ordered by: angel sketch, tiger sketch, angel painting, tiger painting, and panda. We refer readers to the Appendix D.2 for the detailed derivation.

**Main analysis.** Following the same assumption in Sec. 4.3, we are primarily interested in analyzing the difference of the representation space derived from $A$ and $A_1$ and put analysis on the top-3 eigenvectors $\widehat{V}_1 \in \mathbb{R}^{5\times 3}$.

**Theorem B.1.** *Denote $\alpha' = \frac{\alpha}{\rho}$ and $\beta' = \frac{\beta}{\rho}$ and assume $\eta_u = 5, \eta_l = 1$, we have:*

$$\widehat{V}_1 = \begin{bmatrix} \sqrt{2} & \sqrt{2} & 1 & 1 & 1 \\ a(\widehat{\lambda}_2) & a(\widehat{\lambda}_2) & b(\widehat{\lambda}_2) & b(\widehat{\lambda}_2) & 1 \\ c(\widehat{\lambda}_3) & -c(\widehat{\lambda}_3) & -1 & 1 & 0 \end{bmatrix}^\top \cdot R, \quad \mathcal{E}(f_1) = 0, \text{ if } \alpha > 0, \beta > 0. \tag{16}$$

*where $a(\lambda) = \frac{\sqrt{2}(1-6\beta'-\lambda)}{8\beta'}, b(\lambda) = \frac{4\beta'-1+\lambda}{4\beta'}, c(\lambda) = \frac{\sqrt{2}(1-3\alpha'-6\beta'-\lambda)}{3\alpha'}$. $R$ is a diagonal matrix that normalizes the eigenvectors to unit norm and $\widehat{\lambda}_2, \widehat{\lambda}_3$ are the 2nd and 3rd highest eigenvalues.*

**Interpretation.** When semantic OOD shares the same domain as covariate OOD, the OOD generalization error $\mathcal{E}(f_1)$ can be reduced to 0 as long as $\alpha$ and $\beta$ are positive. This generalization ability shows that semantic OOD and covariate OOD sharing the same domain could benefit OOD generalization. We empirically verify our theory in Section E.4.

**Theorem B.2.** *Denote $\alpha' = \frac{\alpha}{\rho}$ and $\beta' = \frac{\beta}{\rho}$ and assume $\eta_u = 5, \eta_l = 1$, we have:*

$$\mathcal{S}(f) - \mathcal{S}(f_1) \begin{cases} > 0 & \text{, if } \alpha', \beta' \in \text{black area in Figure 6 (b)}; \\ < 0 & \text{, if } \alpha', \beta' \in \text{white area in Figure 6 (b)}. \end{cases} \tag{17}$$

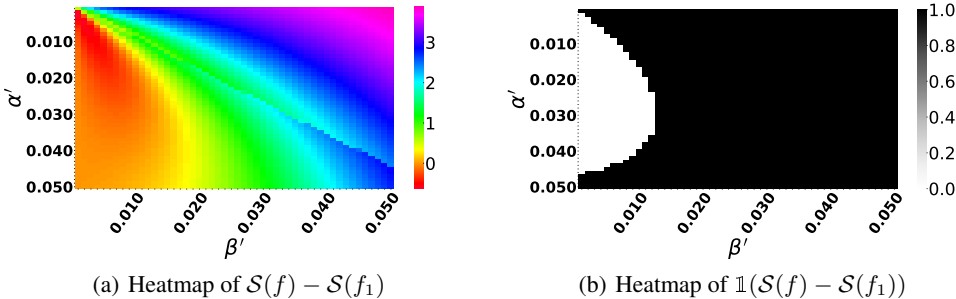

(a) Heatmap of $\mathcal{S}(f) - \mathcal{S}(f_1)$        (b) Heatmap of $\mathbb{1}(\mathcal{S}(f) - \mathcal{S}(f_1))$

Figure 6: Visualization of the separability difference between two cases defined in Figure 5 (a) and Figure 5 (b). Figure 6 (a) utilizes a heatmap to depict the distribution, while Figure 6 (a) uses the indicator function.

**Interpretation.** If $\alpha', \beta' \in$ black area in Figure 6 (b) and semantic OOD comes from a different domain, this would increase the separability between ID and semantic OOD, which benefits OOD detection. If $\alpha', \beta' \in$ white area in Figure 6 (b) and semantic OOD comes from a different domain, this would impair OOD detection.

## C  Impacts of ID Labels on OOD Generalization and Detection

Compared to spectral contrastive loss proposed by Haochen et al. [7], we utilize ID labels in the pre-training. In this section, we analyze the impacts of ID labels on the OOD generalization and detection performance.

Following the same assumption in Sec. 4.3, we are primarily interested in analyzing the difference of the representation space derived from $A$ and $A^{(u)}$ and put analysis on the top-3 eigenvectors $\widehat{V}^{(u)} \in \mathbb{R}^{5 \times 3}$. Detailed derivation can be found in the Appendix D.3.

**Theorem C.1.** *Assume $\eta_u = 5, \eta_l = 1$, we have:*

$$\widehat{V}^{(u)} = \begin{cases} \frac{1}{2} \begin{bmatrix} 1 & 1 & 1 & 1 & 0 \\ 0 & 0 & 0 & 0 & 2 \\ -1 & 1 & -1 & 1 & 0 \end{bmatrix}^{\top} & , \text{if } \alpha > \beta; \\[2em] \frac{1}{2} \begin{bmatrix} 1 & 1 & 1 & 1 & 0 \\ 0 & 0 & 0 & 0 & 2 \\ -1 & -1 & 1 & 1 & 0 \end{bmatrix}^{\top} & , \text{if } \alpha < \beta. \end{cases}, \mathcal{E}(f^{(u)}) = \begin{cases} 0 & , \text{if } \alpha > \beta; \\ 2 & , \text{if } \alpha < \beta. \end{cases} \tag{18}$$

**Interpretation.** By comparing the eigenvectors $\widehat{V}$ in the supervised case (Theorem 4.1) and the eigenvectors $\widehat{V}^{(u)}$ in the self-supervised case, we find that adding ID label information transforms the performance condition from $\alpha = \beta$ to $\frac{9}{8}\alpha = \beta$. In particular, the discussion can be divided into two cases: (1) $\alpha > \beta$. (2) $\alpha < \beta$. In the first case when the connection between the class is stronger than the domain, the model could learn a perfect ID classifier based on features in the first two rows in $\widehat{V}^{(u)}$ and effectively generalize to the covariate-shifted domain (the third and fourth row in $\widehat{V}^{(u)}$), achieving perfect OOD generalization with $\mathcal{E}(f^{(u)}) = 0$. In the second case when the connection between the domain is stronger than the connection between the class, the embeddings of covariate-shifted OOD data are identical, resulting in high OOD generalization error.

**Theorem C.2.** *Assume $\eta_u = 5, \eta_l = 1$, we have:*

$$\mathcal{S}(f) - \mathcal{S}(f^{(u)}) > 0, \text{if } \alpha > 0, \beta > 0 \tag{19}$$

**Interpretation.** After incorporating ID label information, the separability between ID and semantic OOD in the learned embedding space increases as long as $\alpha$ and $\beta$ are positive. This suggests that ID label information indeed helps OOD detection. We empirically verify our theory in Section E.4.

# D Technical Details of Derivation

## D.1 Details for Figure 5 (a)

**Augmentation Transformation Probability**. Recall the augmentation transformation probability, which encodes the probability of augmenting an original image $\bar{x}$ to the augmented view $x$:

$$\mathcal{T}(x \mid \bar{x}) = \begin{cases} \rho & \text{if} \quad y(\bar{x}) = y(x), d(\bar{x}) = d(x); \\ \alpha & \text{if} \quad y(\bar{x}) = y(x), d(\bar{x}) \neq d(x); \\ \beta & \text{if} \quad y(\bar{x}) \neq y(x), d(\bar{x}) = d(x); \\ \gamma & \text{if} \quad y(\bar{x}) \neq y(x), d(\bar{x}) \neq d(x). \end{cases}$$

Thus, the augmentation matrix $\mathcal{T}$ of the toy example shown in Figure 5 (a) can be given by:

$$\mathcal{T} = \begin{bmatrix} \rho & \beta & \alpha & \gamma & \gamma \\ \beta & \rho & \gamma & \alpha & \gamma \\ \alpha & \gamma & \rho & \beta & \gamma \\ \gamma & \alpha & \beta & \rho & \gamma \\ \gamma & \gamma & \gamma & \gamma & \rho \end{bmatrix}$$

Each row or column encodes augmentation connectivity associated with a specific sample, ordered by: angel sketch, tiger sketch, angel painting, tiger painting, and panda.

**Details for $A^{(u)}$ and $A^{(l)}$.** Recall that the self-supervised connectivity is defined in Eq. 1. Since we have a 5-nodes graph, $A^{(u)}$ would be $\frac{1}{5}\mathcal{T}\mathcal{T}^\top$. If we assume $\eta_u = 5$, we can derive the closed-form self-supervised adjacency matrix:

$$\eta_u A^{(u)} = \begin{bmatrix} \rho^2+\beta^2+\alpha^2+2\gamma^2 & 2\rho\beta+\gamma^2+2\gamma\alpha & 2\rho\alpha+\gamma^2+2\gamma\beta & 2\alpha\beta+\gamma^2+2\gamma\rho & \gamma(\gamma+\alpha+\beta+2\rho) \\ 2\rho\beta+\gamma^2+2\gamma\alpha & \rho^2+\beta^2+\alpha^2+2\gamma^2 & 2\alpha\beta+\gamma^2+2\gamma\rho & 2\rho\alpha+\gamma^2+2\gamma\beta & \gamma(\gamma+\alpha+\beta+2\rho) \\ 2\rho\alpha+\gamma^2+2\gamma\beta & 2\alpha\beta+\gamma^2+2\gamma\rho & \rho^2+\beta^2+\alpha^2+2\gamma^2 & 2\rho\beta+\gamma^2+2\gamma\alpha & \gamma(\gamma+\alpha+\beta+2\rho) \\ 2\alpha\beta+\gamma^2+2\gamma\rho & 2\rho\alpha+\gamma^2+2\gamma\beta & 2\rho\beta+\gamma^2+2\gamma\alpha & \rho^2+\beta^2+\alpha^2+2\gamma^2 & \gamma(\gamma+\alpha+\beta+2\rho) \\ \gamma(\gamma+\alpha+\beta+2\rho) & \gamma(\gamma+\alpha+\beta+2\rho) & \gamma(\gamma+\alpha+\beta+2\rho) & \gamma(\gamma+\alpha+\beta+2\rho) & \rho^2+4\gamma^2 \end{bmatrix}$$

Then, according to the supervised connectivity defined in Eq. 2, we only compute ID-labeled data. Since we have two known classes and each class contains one sample, $A^{(l)} = \mathcal{T}_{:,1}\mathcal{T}_{:,1}^\top + \mathcal{T}_{:,2}\mathcal{T}_{:,2}^\top$. Then if we let $\eta_l = 1$, we can have the closed-form supervised adjacency matrix:

$$\eta_l A^{(l)} = \begin{bmatrix} \rho^2+\beta^2 & 2\rho\beta & \rho\alpha+\gamma\beta & \alpha\beta+\gamma\rho & \gamma(\rho+\beta) \\ 2\rho\beta & \rho^2+\beta^2 & \alpha\beta+\gamma\rho & \rho\alpha+\gamma\beta & \gamma(\rho+\beta) \\ \rho\alpha+\gamma\beta & \alpha\beta+\gamma\rho & \alpha^2+\gamma^2 & 2\gamma\alpha & \gamma(\alpha+\gamma) \\ \alpha\beta+\gamma\rho & \rho\alpha+\gamma\beta & 2\gamma\alpha & \alpha^2+\gamma^2 & \gamma(\alpha+\gamma) \\ \gamma(\rho+\beta) & \gamma(\rho+\beta) & \gamma(\alpha+\gamma) & \gamma(\alpha+\gamma) & 2\gamma^2 \end{bmatrix}$$

**Details of eigenvectors $\widehat{V}$.** We assume $\rho \gg \max(\alpha,\beta) \geq \min(\alpha,\beta) \gg \gamma \geq 0$, and denote $\alpha' = \frac{\alpha}{\rho}, \beta' = \frac{\beta}{\rho}$. $A$ can be approximately given by:

$$A \approx \widehat{A} = \frac{1}{\widehat{C}} \begin{bmatrix} 2 & 4\beta' & 3\alpha' & 0 & 0 \\ 4\beta' & 2 & 0 & 3\alpha' & 0 \\ 3\alpha' & 0 & 1 & 2\beta' & 0 \\ 0 & 3\alpha' & 2\beta' & 1 & 0 \\ 0 & 0 & 0 & 0 & 1 \end{bmatrix},$$

where $\widehat{C}$ is the normalization term and equals to $7 + 12\beta' + 12\alpha'$. The squares of the minimal term (e.g., $\frac{\alpha\beta}{\rho^2}, \frac{\alpha^2}{\rho^2}, \frac{\beta^2}{\rho^2}, \frac{\gamma}{\rho} = \frac{\gamma}{\alpha} \cdot \frac{\alpha}{\rho}, \frac{\alpha\gamma}{\rho^2}$, etc) are approximated to 0.

$$\widehat{D} = \frac{1}{\widehat{C}}\text{diag}[2+4\beta'+3\alpha', 2+4\beta'+3\alpha', 1+2\beta'+3\alpha', 1+2\beta'+3\alpha', 1]$$

$$\widehat{D^{-\frac{1}{2}}} = \sqrt{\widehat{C}}\,\text{diag}[\frac{1}{\sqrt{2}}(1-\beta'-\frac{3}{4}\alpha'), \frac{1}{\sqrt{2}}(1-\beta'-\frac{3}{4}\alpha'), 1-\beta'-\frac{3}{2}\alpha', 1-\beta'-\frac{3}{2}\alpha', 1]$$

$$D^{-\frac{1}{2}}AD^{-\frac{1}{2}} \approx \widehat{D^{-\frac{1}{2}}}\widehat{A}\widehat{D^{-\frac{1}{2}}} = \begin{bmatrix} 1-2\beta'-\frac{3}{2}\alpha' & 2\beta' & \frac{3}{\sqrt{2}}\alpha' & 0 & 0 \\ 2\beta' & 1-2\beta'-\frac{3}{2}\alpha' & 0 & \frac{3}{\sqrt{2}}\alpha' & 0 \\ \frac{3}{\sqrt{2}}\alpha' & 0 & 1-2\beta'-3\alpha' & 2\beta' & 0 \\ 0 & \frac{3}{\sqrt{2}}\alpha' & 2\beta' & 1-2\beta'-3\alpha' & 0 \\ 0 & 0 & 0 & 0 & 1 \end{bmatrix}$$

Let $\lambda_{1,\ldots,5}$ and $v_{1,\ldots,5}$ be the eigenvalues and their corresponding eigenvectors of $D^{-\frac{1}{2}}AD^{-\frac{1}{2}}$. Then the concrete form of $\lambda_{1,\ldots,5}$ and $v_{1,\ldots,5}$ can be approximately given by:

$$
\begin{aligned}
\widehat{v}_1 &= \tfrac{1}{\sqrt{6}}[\sqrt{2}, \sqrt{2}, 1, 1, 0]^\top & \widehat{\lambda}_1 &= 1 \\
\widehat{v}_2 &= [0, 0, 0, 0, 1]^\top & \widehat{\lambda}_2 &= 1 \\
\widehat{v}_3 &= \tfrac{1}{\sqrt{6}}[-\sqrt{2}, \sqrt{2}, -1, 1, 0]^\top & \widehat{\lambda}_3 &= 1 - 4\beta' \\
\widehat{v}_4 &= \tfrac{1}{\sqrt{6}}[-1, -1, \sqrt{2}, \sqrt{2}, 0]^\top & \widehat{\lambda}_4 &= 1 - \tfrac{9}{2}\alpha' \\
\widehat{v}_5 &= \tfrac{1}{\sqrt{6}}[1, -1, -\sqrt{2}, \sqrt{2}, 0]^\top & \widehat{\lambda}_5 &= 1 - 4\beta' - \tfrac{9}{2}\alpha'
\end{aligned}
$$

Since $\alpha', \beta' > 0$, we can always have $\widehat{\lambda}_1 = \widehat{\lambda}_2 > \widehat{\lambda}_3 > \widehat{\lambda}_5$ and $\widehat{\lambda}_1 = \widehat{\lambda}_2 > \widehat{\lambda}_4 > \widehat{\lambda}_5$. Then, we let $k = 3$ and $\widehat{V} \in \mathbb{R}^{5\times3}$ is given by:

$$
\widehat{V} = \begin{cases}
\begin{bmatrix} \frac{1}{\sqrt{3}} & \frac{1}{\sqrt{3}} & \frac{1}{\sqrt{6}} & \frac{1}{\sqrt{6}} & 0 \\ 0 & 0 & 0 & 0 & 1 \\ -\frac{1}{\sqrt{3}} & \frac{1}{\sqrt{3}} & -\frac{1}{\sqrt{6}} & \frac{1}{\sqrt{6}} & 0 \end{bmatrix}^\top & \text{, if } \frac{9}{8}\alpha' > \beta'; \\[3em]
\begin{bmatrix} \frac{1}{\sqrt{3}} & \frac{1}{\sqrt{3}} & \frac{1}{\sqrt{6}} & \frac{1}{\sqrt{6}} & 0 \\ 0 & 0 & 0 & 0 & 1 \\ -\frac{1}{\sqrt{6}} & -\frac{1}{\sqrt{6}} & \frac{1}{\sqrt{3}} & \frac{1}{\sqrt{3}} & 0 \end{bmatrix}^\top & \text{, if } \frac{9}{8}\alpha' < \beta'.
\end{cases}
$$

**Details of linear probing and separability evaluation.** Recall that the closed-form embedding $Z = [D]^{-\frac{1}{2}}V_k\sqrt{\Sigma_k}$. Based on the derivation above, closed-form features for ID sample $Z_{\text{in}} \in \mathbb{R}^{2\times3}$ can be approximately given by:

$$
\widehat{Z}_{\text{in}} = \begin{cases}
\frac{(1-\beta'-0.75\alpha')\sqrt{\widehat{C}}}{\sqrt{6}} \begin{bmatrix} 1 & 0 & -\sqrt{1-4\beta'} \\ 1 & 0 & \sqrt{1-4\beta'} \end{bmatrix} & \text{, if } \frac{9}{8}\alpha' > \beta'. \\[2em]
\frac{(1-\beta'-0.75\alpha')\sqrt{\widehat{C}}}{2\sqrt{3}} \begin{bmatrix} \sqrt{2} & 0 & -\sqrt{1-\frac{9}{2}\alpha'} \\ \sqrt{2} & 0 & -\sqrt{1-\frac{9}{2}\alpha'} \end{bmatrix} & \text{, if } \frac{9}{8}\alpha' < \beta'.
\end{cases}
$$

Based on the least error method, we can derive the weights of the linear classifier $M \in \mathbb{R}^{3\times2}$,

$$
\widehat{M} = (\widehat{Z}_{\text{in}}^\top \widehat{Z}_{\text{in}})^\dagger \widehat{Z}_{\text{in}}^T y_{\text{in}}
$$

where $(\cdot)^\dagger$ is the Moore-Penrose inverse and $y_{\text{in}}$ is the one-hot encoded ground truth class labels. So when $\frac{9}{8}\alpha > \beta$, the predicted probability $\widehat{y}_{\text{covariate}}$ can be given by:

$$
\widehat{y}_{\text{out}}^{\text{covariate}} = \hat{Z}_{\text{out}}^{\text{covariate}} \cdot \hat{M} = \frac{(1-\beta'-\frac{3}{2}\alpha')}{1-\beta'-\frac{3}{4}\alpha'} \cdot \mathcal{I}
$$

where $\mathcal{I} \in \mathbb{R}^{2\times2}$ is an identity matrix. We notice that when $\frac{9}{8}\alpha < \beta$, the closed-form features for ID samples are identical, indicating the impossibility of learning a clear boundary to classify classes angel and tiger. Eventually, we can derive the linear probing error:

$$
\mathcal{E}(f) = \begin{cases} 0 & \text{, if } \frac{9}{8}\alpha > \beta; \\[1em] 2 & \text{, if } \frac{9}{8}\alpha < \beta. \end{cases}
$$

The separability between ID data and semantic OOD data can be computed based on the closed-form embeddings $\widehat{Z}_{\text{in}}$ and $\widehat{Z}_{\text{out}}^{\text{semantic}}$:

$$
\widehat{Z}_{\text{out}}^{\text{semantic}} = \sqrt{\widehat{C}} \cdot [0, 1, 0]
$$

$$
\mathcal{S}(f) = \begin{cases} (7 + 12\beta' + 12\alpha')(\frac{1-2\beta'}{3}(1 - \beta' - \frac{3}{4}\alpha')^2 + 1) & \text{, if } \frac{9}{8}\alpha > \beta; \\[1em] (7 + 12\beta' + 12\alpha')(\frac{2-3\alpha'}{8}(1 - \beta' - \frac{3}{4}\alpha')^2 + 1) & \text{, if } \frac{9}{8}\alpha < \beta. \end{cases}
$$

## D.2 Details for Figure 5 (b)

**Augmentation Transformation Probability.** Illustrated in Figure 5 (b), when semantic OOD and covariate OOD share the same domain, the augmentation matrix can be slightly different from the previous case:

$$
\mathcal{T} = \begin{bmatrix}
\rho & \beta & \alpha & \gamma & \gamma \\
\beta & \rho & \gamma & \alpha & \gamma \\
\alpha & \gamma & \rho & \beta & \beta \\
\gamma & \alpha & \beta & \rho & \beta \\
\gamma & \gamma & \beta & \beta & \rho
\end{bmatrix}
$$

Each row or column represents augmentation connectivity of a specific sample, ordered by: angel sketch, tiger sketch, angel painting, tiger painting, and panda.

**Details for $A_1^{(u)}$ and $A_1^{(l)}$.** After the assumption $\eta_u = 5, \eta_l = 1$, we can have $\eta_u A_1^{(u)} = \mathcal{T}\mathcal{T}^\top$:

$$
\eta_u A_1^{(u)} = \begin{bmatrix}
\rho^2 + \beta^2 + \alpha^2 + 2\gamma^2 & 2\rho\beta + \gamma^2 + 2\gamma\alpha & 2\rho\alpha + 3\gamma\beta & 2\alpha\beta + \gamma\beta + 2\gamma\rho & \alpha\beta + 2\gamma(\beta + \rho) \\
2\rho\beta + \gamma^2 + 2\gamma\alpha & \rho^2 + \beta^2 + \alpha^2 + 2\gamma^2 & 2\alpha\beta + \gamma\beta + 2\gamma\rho & 2\rho\alpha + 3\gamma\beta & \alpha\beta + 2\gamma(\beta + \rho) \\
2\rho\alpha + 3\gamma\beta & 2\alpha\beta + \gamma\beta + 2\gamma\rho & \rho^2 + 2\beta^2 + \alpha^2 + \gamma^2 & 2\rho\beta + \beta^2 + 2\gamma\alpha & 2\rho\beta + \beta^2 + \gamma^2 + \gamma\alpha \\
2\alpha\beta + \gamma\beta + 2\gamma\rho & 2\alpha\rho + 3\gamma\beta & 2\rho\beta + \beta^2 + 2\gamma\alpha & \rho^2 + 2\beta^2 + \alpha^2 + \gamma^2 & 2\rho\beta + \beta^2 + \gamma^2 + \gamma\alpha \\
\alpha\beta + 2\gamma(\beta + \rho) & \alpha\beta + 2\gamma(\beta + \rho) & 2\rho\beta + \beta^2 + \gamma^2 + \gamma\alpha & 2\rho\beta + \beta^2 + \gamma^2 + \gamma\alpha & \rho^2 + 2\beta^2 + 2\gamma^2
\end{bmatrix}
$$

And the supervised adjacency matrix $A_1^{(l)} = \mathcal{T}_{:,1}\mathcal{T}_{:,1}^\top + \mathcal{T}_{:,2}\mathcal{T}_{:,2}^\top$ can be given by:

$$
\eta_l A_1^{(l)} = \begin{bmatrix}
\rho^2 + \beta^2 & 2\rho\beta & \rho\alpha + \gamma\beta & \alpha\beta + \gamma\rho & \gamma(\beta + \rho) \\
2\rho\beta & \rho^2 + \beta^2 & \alpha\beta + \gamma\rho & \rho\alpha + \gamma\beta & \gamma(\beta + \rho) \\
\rho\alpha + \gamma\beta & \alpha\beta + \gamma\rho & \alpha^2 + \gamma^2 & 2\gamma\alpha & \gamma(\gamma + \alpha) \\
\alpha\beta + \gamma\rho & \rho\alpha + \gamma\beta & 2\gamma\alpha & \alpha^2 + \gamma^2 & \gamma(\gamma + \alpha) \\
\gamma(\beta + \rho) & \gamma(\beta + \rho) & \gamma(\gamma + \alpha) & \gamma(\gamma + \alpha) & 2\gamma^2
\end{bmatrix}
$$

**Details for $\widehat{V}_1$.** Following the same assumption, the adjacency matrix can be approximately given by:

$$
A_1 \approx \widehat{A}_1 = \frac{1}{\widehat{C}_1} \begin{bmatrix}
2 & 4\beta' & 3\alpha' & 0 & 0 \\
4\beta' & 2 & 0 & 3\alpha' & 0 \\
3\alpha' & 0 & 1 & 2\beta' & 2\beta' \\
0 & 3\alpha' & 2\beta' & 1 & 2\beta' \\
0 & 0 & 2\beta' & 2\beta' & 1
\end{bmatrix}
$$

$$
\widehat{D}_1 = \frac{1}{\widehat{C}_1} \cdot \mathrm{diag}[2 + 4\beta' + 3\alpha', 2 + 4\beta' + 3\alpha', 1 + 4\beta' + 3\alpha', 1 + 4\beta' + 3\alpha', 1 + 4\beta']
$$

$$
\widehat{D_1^{-\frac{1}{2}}} = \sqrt{\widehat{C}_1} \cdot \mathrm{diag}[\frac{1}{\sqrt{2}}(1 - \beta' - \frac{3}{4}\alpha'), \frac{1}{\sqrt{2}}(1 - \beta' - \frac{3}{4}\alpha'), 1 - 2\beta' - \frac{3}{2}\alpha', 1 - 2\beta' - \frac{3}{2}\alpha', 1 - 2\beta']
$$

$$
D_1^{-\frac{1}{2}} A_1 D_1^{-\frac{1}{2}} \approx \widehat{D_1^{-\frac{1}{2}}} \widehat{A}_1 \widehat{D_1^{-\frac{1}{2}}} = \begin{bmatrix}
1 - 2\beta' - \frac{3}{2}\alpha' & 2\beta' & \frac{3}{\sqrt{2}}\alpha' & 0 & 0 \\
2\beta' & 1 - 2\beta' - \frac{3}{2}\alpha' & 0 & \frac{3}{\sqrt{2}}\alpha' & 0 \\
\frac{3}{\sqrt{2}}\alpha' & 0 & 1 - 4\beta' - 3\alpha' & 2\beta' & 2\beta' \\
0 & \frac{3}{\sqrt{2}}\alpha' & 2\beta' & 1 - 4\beta' - 3\alpha' & 2\beta' \\
0 & 0 & 2\beta' & 2\beta' & 1 - 4\beta'
\end{bmatrix}
$$

where $\widehat{C}_1$ is the normalization term and $\widehat{C}_1 = 7 + 20\beta' + 12\alpha'$. After eigendecomposition, we can derive ordered eigenvalues and their corresponding eigenvectors:

$\widehat{v}_1 = \frac{1}{\sqrt{7}}[\sqrt{2}, \sqrt{2}, 1, 1, 1]^\top$ $\qquad$ $\widehat{\lambda}_1 = 1$

$\widehat{v}_2 = \frac{1}{\sqrt{2a(\widehat{\lambda}_2)^2 + 2b(\widehat{\lambda}_2)^2 + 1}}[a(\widehat{\lambda}_2), a(\widehat{\lambda}_2), b(\widehat{\lambda}_2), b(\widehat{\lambda}_2), 1]^\top$ $\qquad$ $\widehat{\lambda}_2 = 1 - 3b + \frac{\sqrt{3}\sqrt{(27a^2 - 40ab + 48b^2)} - 9a}{4}$

$\widehat{v}_3 = \frac{1}{\sqrt{2c(\widehat{\lambda}_3)^2 + 2}}[c(\widehat{\lambda}_3), -c(\widehat{\lambda}_3), -1, 1, 0]^\top$ $\qquad$ $\widehat{\lambda}_3 = 1 - 5b + \frac{\sqrt{81a^2 + 24ab + 16b^2} - 9a}{4}$

$\widehat{v}_4 = \frac{1}{\sqrt{2a(\widehat{\lambda}_4)^2 + 2b(\widehat{\lambda}_4)^2 + 1}}[a(\widehat{\lambda}_4), a(\widehat{\lambda}_4), b(\widehat{\lambda}_4), b(\widehat{\lambda}_4), 1]^\top$ $\qquad$ $\widehat{\lambda}_4 = 1 - 3b - \frac{\sqrt{3}\sqrt{(27a^2 - 40ab + 48b^2)} + 9a}{4}$

$\widehat{v}_5 = \frac{1}{\sqrt{2c(\widehat{\lambda}_5)^2 + 2}}[c(\widehat{\lambda}_5), -c(\widehat{\lambda}_5), -1, 1, 0]^\top,$ $\qquad$ $\widehat{\lambda}_5 = 1 - 5b - \frac{\sqrt{81a^2 + 24ab + 16b^2} + 9a}{4}$

where $\widehat{\lambda}_1 > \widehat{\lambda}_2 > \widehat{\lambda}_3 > \widehat{\lambda}_4 > \widehat{\lambda}_5$ and $a(\lambda) = \frac{\sqrt{2}(1-6\beta'-\lambda)}{8\beta'}, b(\lambda) = \frac{4\beta'-1+\lambda}{4\beta'}, c(\lambda) = \frac{\sqrt{2}(1-3\alpha'-6\beta'-\lambda)}{3\alpha'}$. We can get closed-form eigenvectors:

$$
\widehat{V}_1 = \begin{bmatrix} \sqrt{2} & \sqrt{2} & 1 & 1 & 1 \\ a(\widehat{\lambda}_2) & a(\widehat{\lambda}_2) & b(\widehat{\lambda}_2) & b(\widehat{\lambda}_2) & 1 \\ c(\widehat{\lambda}_3) & -c(\widehat{\lambda}_3) & -1 & 1 & 0 \end{bmatrix}^{\top} \cdot \operatorname{diag}\left[\frac{1}{\sqrt{7}}, \frac{1}{\sqrt{2a(\widehat{\lambda}_2)^2 + 2b(\widehat{\lambda}_2)^2 + 1}}, \frac{1}{\sqrt{2c(\widehat{\lambda}_3)^2 + 2}}\right]
$$

**Details for linear probing and separability evaluation.** Following the same derivation, we can derive closed-form embedding for ID samples $\widehat{Z}_{\text{in}} = \widehat{D_{\text{in}}^{-\frac{1}{2}}} \widehat{V}_{\text{in}} \sqrt{\widehat{\Sigma}_{\text{in}}}$ and the linear layer weights $\widehat{M} = (\widehat{Z}_{\text{in}}^{\top} \widehat{Z}_{\text{in}})^{\dagger} \widehat{Z}_{\text{in}}^{T} y_{\text{in}}$. Eventually, we can derive the approximately predicted probability $\hat{y}_{\text{out}}^{\text{covariate}}$:

$$
\hat{y}_{\text{out}}^{\text{covariate}} = \begin{bmatrix} a_1 + b_1 & a_1 - b_1 \\ a_1 - b_1 & a_1 + b_1 \end{bmatrix}
$$

where $a_1, b_1 \in \mathbb{R}$ and $b_1 > 0$. This indicates that linear probing error $\mathcal{E}(f_1) = 0$ as long as $\alpha$ and $\beta$ are positive.

Having obtained closed-form representation $Z_{\text{in}}$ and $Z_{\text{out}}^{\text{semantic}}$, we can compute separability $S(f_1)$ and then prove:

$$
\widehat{Z}_{\text{in}} = \frac{(1 - \beta' - \frac{3}{4}\alpha')\sqrt{\widehat{C}_1}}{\sqrt{2}} \begin{bmatrix} \frac{\sqrt{2}}{\sqrt{7}} & \frac{a(\widehat{\lambda}_2)\sqrt{\widehat{\lambda}_2}}{\sqrt{2a(\widehat{\lambda}_2)^2 + 2b(\widehat{\lambda}_2)^2 + 1}} & -\frac{c(\widehat{\lambda}_3)\sqrt{\widehat{\lambda}_3}}{\sqrt{2c(\widehat{\lambda}_3)^2 + 2}} \\ \frac{\sqrt{2}}{\sqrt{7}} & \frac{a(\widehat{\lambda}_2)\sqrt{\widehat{\lambda}_2}}{\sqrt{2a(\widehat{\lambda}_2)^2 + 2b(\widehat{\lambda}_2)^2 + 1}} & \frac{c(\widehat{\lambda}_3)\sqrt{\widehat{\lambda}_3}}{\sqrt{2c(\widehat{\lambda}_3)^2 + 2}} \end{bmatrix}
$$

$$
\widehat{Z}_{\text{out}}^{\text{semantic}} = (1 - 2\beta')\sqrt{\widehat{C}_1}\left[\frac{1}{\sqrt{7}}, \frac{\sqrt{\widehat{\lambda}_2}}{\sqrt{2a(\widehat{\lambda}_2)^2 + 2b(\widehat{\lambda}_2)^2 + 1}}, 0\right]
$$

$$
\mathcal{S}(f) - \mathcal{S}(f_1) \begin{cases} > 0 & \text{, if } \alpha', \beta' \in \text{black area in Figure 6 (b);} \\ < 0 & \text{, if } \alpha', \beta' \in \text{white area in Figure 6 (b).} \end{cases}
$$

### D.3 Calculation Details for self-supervised case

Our analysis for the self-supervised case is based on Figure 5 (a), the adjacency matrix is exactly the same as Eq. 10. After approximation, we can derive:

$$
A^{(u)} \approx \widehat{A}^{(u)} = \frac{1}{\widehat{C}^{(u)}} \begin{bmatrix} 1 & 2\beta' & 2\alpha' & 0 & 0 \\ 2\beta' & 1 & 0 & 2\alpha' & 0 \\ 2\alpha' & 0 & 1 & 2\beta' & 0 \\ 0 & 2\alpha' & 2\beta' & 1 & 0 \\ 0 & 0 & 0 & 0 & 1 \end{bmatrix}
$$

$$
\widehat{D^{(u)}}^{-\frac{1}{2}} = \sqrt{5 + 8\beta' + 8\alpha'} \cdot \operatorname{diag}[1 - \beta' - \alpha', 1 - \beta' - \alpha', 1 - \beta' - \alpha', 1 - \beta' - \alpha', 1]
$$

$$
\widehat{D^{(u)}}^{-\frac{1}{2}} \widehat{A^{(u)}} \widehat{D^{(u)}}^{-\frac{1}{2}} = \begin{bmatrix} 1 - 2\beta' - 2\alpha' & 2\beta' & 2\alpha' & 0 & 0 \\ 2\beta' & 1 - 2\beta' - 2\alpha' & 0 & 2\alpha' & 0 \\ 2\alpha' & 0 & 1 - 2\beta' - 2\alpha' & 2\beta' & 0 \\ 0 & 2\alpha' & 2\beta' & 1 - 2\beta' - 2\alpha' & 0 \\ 0 & 0 & 0 & 0 & 1 \end{bmatrix}
$$

$$
\begin{aligned}
\widehat{v}_1 &= \tfrac{1}{2}[1, 1, 1, 1, 0]^{\top} & \widehat{\lambda}_1 &= 1 \\
\widehat{v}_2 &= [0, 0, 0, 0, 1]^{\top} & \widehat{\lambda}_2 &= 1 \\
\widehat{v}_3 &= \tfrac{1}{2}[-1, 1, -1, 1, 0]^{\top} & \widehat{\lambda}_3 &= 1 - 4\beta' \\
\widehat{v}_4 &= \tfrac{1}{2}[-1, -1, 1, 1, 0]^{\top} & \widehat{\lambda}_4 &= 1 - 4\alpha' \\
\widehat{v}_5 &= \tfrac{1}{2}[1, -1, -1, 1, 0]^{\top} & \widehat{\lambda}_5 &= 1 - 4\alpha' - 4\beta'
\end{aligned}
$$

Following the same procedure presented above, we can prove Theorem C.1 and C.2.

# E More Experiments

## E.1 Dataset Statistics

We provide a detailed description of the datasets used in this work below:

**CIFAR-10** [14] contains $60,000$ color images with 10 classes. The training set has $50,000$ images and the test set has $10,000$ images.

**ImageNet-100** consists of a subset of 100 categories from ImageNet-1K [109]. This dataset contains the following classes: n01498041, n01514859, n01582220, n01608432, n01616318, n01687978, n01776313, n01806567, n01833805, n01882714, n01910747, n01944390, n01985128, n02007558, n02071294, n02085620, n02114855, n02123045, n02128385, n02129165, n02129604, n02165456, n02190166, n02219486, n02226429, n02279972, n02317335, n02326432, n02342885, n02363005, n02391049, n02395406, n02403003, n02422699, n02442845, n02444819, n02480855, n02510455, n02640242, n02672831, n02687172, n02701002, n02730930, n02769748, n02782093, n02787622, n02793495, n02799071, n02802426, n02814860, n02840245, n02906734, n02948072, n02980441, n02999410, n03014705, n03028079, n03032252, n03125729, n03160309, n03179701, n03220513, n03249569, n03291819, n03384352, n03388043, n03450230, n03481172, n03594734, n03594945, n03627232, n03642806, n03649909, n03661043, n03676483, n03724870, n03733281, n03759954, n03761084, n03773504, n03804744, n03916031, n03938244, n04004767, n04026417, n04090263, n04133789, n04153751, n04296562, n04330267, n04371774, n04404412, n04465501, n04485082, n04507155, n04536866, n04579432, n04606251, n07714990, n07745940.

**CIFAR-10-C** is generated based on Hendrycks et al. [15], applying different corruptions on CIFAR-10 including gaussian noise, defocus blur, glass blur, impulse noise, shot noise, snow, and zoom blur.

**ImageNet-100-C** is generated with Gaussian noise added to ImageNet-100 dataset [109].

**SVHN** [16] is a real-world image dataset obtained from house numbers in Google Street View images. This dataset $73,257$ samples for training, and $26,032$ samples for testing with 10 classes.

**Places365** [18] contains scene photographs and diverse types of environments encountered in the world. The scene semantic categories consist of three macro-classes: Indoor, Nature, and Urban.

**LSUN-C** [17] and **LSUN-R** [17] are large-scale image datasets that are annotated using deep learning with humans in the loop. LSUN-C is a cropped version of LSUN and LSUN-R is a resized version of the LSUN dataset.

**Textures** [19] refers to the Describable Textures Dataset, which contains a large dataset of visual attributes including patterns and textures. The subset we used has no overlap categories with the CIFAR dataset [14].

**iNaturalist** [110] is a challenging real-world dataset with iNaturalist species, captured in a wide variety of situations. It has 13 super-categories and 5,089 sub-categories. We use the subset from Huang et al. [111] that contains 110 plant classes that no category overlaps with IMAGENET-1K [109].

**Office-Home** [20] is a challenging dataset, which consists of 15500 images from 65 categories. It is made up of 4 domains: Artistic (Ar), Clip-Art (Cl), Product (Pr), and Real-World (Rw).

**Details of data split for OOD datasets.** For datasets with standard train-test split (e.g., SVHN), we use the original test split for evaluation. For other OOD datasets (e.g., LSUN-C), we use 70% of the data for creating the wild mixture training data as well as the mixture validation dataset. We use the remaining examples for test-time evaluation. For splitting training/validation, we use 30% for validation and the remaining for training. During validation, we could only access unlabeled wild data and labeled clean ID data, which means hyper-parameters are chosen based on the performance of ID Acc. on the ID validation set (more in Section F).

## E.2 Results on ImageNet-100

In this section, we present results on the large-scale dataset ImageNet-100 to further demonstrate our empirical competitive performance. We employ ImageNet-100 as $\mathbb{P}_{in}$, ImageNet-100-C as $\mathbb{P}_{out}^{covariate}$, and iNaturalist [110] as $\mathbb{P}_{out}^{semantic}$. Similar to our CIFAR experiment, we divide the ImageNet-100

| Model | Places365 $\mathbb{P}_{out}^{semantic}$, CIFAR-10-C $\mathbb{P}_{out}^{covariate}$ | | | | LSUN-R $\mathbb{P}_{out}^{semantic}$, CIFAR-10-C $\mathbb{P}_{out}^{covariate}$ | | | |
|---|---|---|---|---|---|---|---|---|
| | OOD Acc.↑ | ID Acc.↑ | FPR↓ | AUROC↑ | OOD Acc.↑ | ID Acc.↑ | FPR↓ | AUROC↑ |
| *OOD detection* | | | | | | | | |
| **MSP** [24] | 75.05 | 94.84 | 57.40 | 84.49 | 75.05 | 94.84 | 52.15 | 91.37 |
| **ODIN** [25] | 75.05 | 94.84 | 57.40 | 84.49 | 75.05 | 94.84 | 26.62 | 94.57 |
| **Energy** [26] | 75.05 | 94.84 | 40.14 | 89.89 | 75.05 | 94.84 | 27.58 | 94.24 |
| **Mahalanobis** [27] | 75.05 | 94.84 | 68.57 | 84.61 | 75.05 | 94.84 | 42.62 | 93.23 |
| **ViM** [28] | 75.05 | 94.84 | **21.95** | **95.48** | 75.05 | 94.84 | 36.80 | 93.37 |
| **KNN** [23] | 75.05 | 94.84 | 42.67 | 91.07 | 75.05 | 94.84 | 29.75 | 94.60 |
| **ASH** [29] | 75.05 | 94.84 | 44.07 | 88.84 | 75.05 | 94.84 | 22.07 | 95.61 |
| *OOD generalization* | | | | | | | | |
| **ERM** [30] | 75.05 | 94.84 | 40.14 | 89.89 | 75.05 | 94.84 | 27.58 | 94.24 |
| **IRM** [31] | 77.92 | 90.85 | 53.79 | 88.15 | 77.92 | 90.85 | 34.50 | 94.54 |
| **Mixup** [32] | 79.17 | 93.30 | 58.24 | 75.70 | 79.17 | 93.30 | 32.73 | 88.86 |
| **VREx** [33] | 76.90 | 91.35 | 56.13 | 87.45 | 76.90 | 91.35 | 44.20 | 92.55 |
| **EQRM** [34] | 75.71 | 92.93 | 51.00 | 88.61 | 75.71 | 92.93 | 31.23 | 94.94 |
| **SharpDRO** [35] | 79.03 | 94.91 | 34.64 | 91.96 | 79.03 | 94.91 | 13.27 | 97.44 |
| *Learning w. $\mathbb{P}_{wild}$* | | | | | | | | |
| **OE** [36] | 35.98 | 94.75 | 27.02 | 94.57 | 46.89 | 94.07 | 0.70 | 99.78 |
| **Energy (w/ outlier)** [26] | 19.86 | 90.55 | 23.89 | 93.60 | 32.91 | 93.01 | 0.27 | 99.94 |
| **Woods** [37] | 54.58 | 94.88 | 30.48 | 93.28 | 78.75 | **95.01** | 0.60 | 99.87 |
| **Scone** [5] | 85.21 | 94.59 | 37.56 | 90.90 | **80.31** | 94.97 | 0.87 | 99.79 |
| **Ours** | **87.04**$_{\pm0.3}$ | 93.40$_{\pm0.3}$ | 40.97$_{\pm1.1}$ | 91.82$_{\pm0.0}$ | 79.38$_{\pm0.8}$ | 92.44$_{\pm0.1}$ | **0.06**$_{\pm0.0}$ | **99.99**$_{\pm0.0}$ |

Table 3: Additional results: comparison with competitive OOD generalization and OOD detection methods on CIFAR-10. To facilitate a fair comparison, we include results from Scone [5] and set $\pi_c = 0.5$, $\pi_s = 0.1$ by default for the mixture distribution $\mathbb{P}_{wild} := (1-\pi_s-\pi_c)\mathbb{P}_{in}+\pi_s\mathbb{P}_{out}^{semantic}+\pi_c\mathbb{P}_{out}^{covariate}$. **Bold**=best. (*Since all the OOD detection methods use the same model trained with the CE loss on $\mathbb{P}_{in}$, they display the same ID and OOD accuracy on CIFAR-10-C.)

training set into 50% labeled as ID and 50% unlabeled. Then we mix unlabeled ImageNet-100, ImageNet-100-C, and iNaturalist to generate the wild dataset. We include results from Scone [5] and set $\pi_c = 0.5$, $\pi_s = 0.1$ for consistency. We pre-train the backbone ResNet-34 [112] with spectral contrastive loss and then use ID data to fine-tune the model. We set the pre-training epoch as 100, batch size as 512, and learning rate as 0.01. For fine-tuning, we set the learning rate to 0.01, batch size to 128, and train for 10 epochs. Empirical results in Table 4 indicate that our method effectively balances OOD generalization and detection while achieving strong performance in both aspects. While Wood [37] displays strong OOD detection performance, the OOD generation performance (44.46%) is significantly worse than ours (72.58%). More detailed implementation can be found in Appendix F.

| Method | OOD Acc.↑ | ID Acc.↑ | FPR↓ | AUROC↑ |
|---|---|---|---|---|
| **Woods** [37] | 44.46 | 86.49 | **10.50** | **98.22** |
| **Scone** [5] | 65.34 | **87.64** | 27.13 | 95.66 |
| **Ours** | **72.58** | 86.68 | 21.00 | 96.52 |

Table 4: Results on ImageNet-100. We employ ImageNet-100 as $\mathbb{P}_{in}$, ImageNet-100-C with Gaussian noise as $\mathbb{P}_{out}^{covariate}$, and iNaturalist as $\mathbb{P}_{out}^{semantic}$. **Bold**=Best.

### E.3 Results on Office-Home

In this section, we present empirical results on the Office-Home [20], a dataset comprising 65 object classes distributed across 4 different domains: Artistic (Ar), Clipart (Cl), Product (Pr), and Real-World (Rw). Following OSBP [113], we separate 65 object classes into the first 25 classes in alphabetic order as ID classes and the remainder of classes as semantic OOD classes. Subsequently, we construct the ID data from one domain (e.g., Ar) across 25 classes, and the covariate OOD from another domain (e.g., Cl) to carry out the OOD generalization task (e.g., Ar → Cl). The semantic OOD data are from the remainder of classes, in the same domain as covariate OOD data. We consider the following wild data, where $\mathbb{P}_{wild} = \pi_c\mathbb{P}_{out}^{covariate} + \pi_s\mathbb{P}_{out}^{semantic}$ and $\pi_c + \pi_s = 1$. This setting is also known as open-set domain adaptation [114], which can be viewed as a special case of ours.

For a fair empirical comparison, we include results from Anna [115], containing comprehensive baselines like STA [116], OSBP [113], DAOD [117], OSLPP [118], ROS [119], and Anna [115].

Following previous literature, we use OOD Acc. to denote the average class accuracy over known classes only in this section. We employ ResNet-50 [112] as the default backbone. As shown in Table 5, our approach strikes a balance between OOD generalization and detection, even outperforming the state-of-the-art method Anna in terms of FPR by 11.3% on average. This demonstrates the effectiveness of our method in handling the complex OOD scenarios present in the Office-Home dataset. More detailed implementation can be found in Appendix F.

| Method | $Ar \to Cl$ | | $Ar \to Pr$ | | $Ar \to Rw$ | | $Cl \to Ar$ | | $Cl \to Pr$ | | $Cl \to Rw$ | | $Pr \to Ar$ | |
|---|---|---|---|---|---|---|---|---|---|---|---|---|---|---|
| | OOD Acc.↑ | FPR↓ | OOD Acc.↑ | FPR↓ | OOD Acc.↑ | FPR↓ | OOD Acc.↑ | FPR↓ | OOD Acc.↑ | FPR↓ | OOD Acc.↑ | FPR↓ | OOD Acc.↑ | FPR↓ |
| STA$_{sum}$ [116] | 50.8 | 36.6 | 68.7 | 40.3 | **81.1** | 49.5 | 53.0 | 36.1 | 61.4 | 36.5 | 69.8 | 36.8 | 55.4 | 26.3 |
| STA$_{max}$ [116] | 46.0 | 27.7 | 68.0 | 51.6 | 78.6 | 39.6 | 51.4 | 35.0 | 61.8 | 40.9 | 67.0 | 33.3 | 54.2 | 27.6 |
| OSBP [113] | 50.2 | 38.9 | 71.8 | 40.2 | 79.3 | 32.5 | **59.4** | 29.7 | 67.0 | 37.3 | 72.0 | 30.8 | 59.1 | 31.9 |
| DAOD [117] | **72.6** | 48.2 | 55.3 | 42.1 | 78.2 | 37.4 | 59.1 | 38.3 | **70.8** | 47.4 | **77.8** | 43.0 | **71.3** | 49.5 |
| OSLPP [118] | 55.9 | 32.9 | **72.5** | 26.9 | 80.1 | 30.6 | 49.6 | 21.0 | 61.6 | 26.7 | 67.2 | 26.1 | 54.6 | 23.8 |
| ROS [119] | 50.6 | 25.9 | 68.4 | 29.7 | 75.8 | 22.8 | 53.6 | 34.5 | 59.8 | 28.4 | 65.3 | 27.8 | 57.3 | 35.7 |
| Anna [115] | 61.4 | 21.3 | 68.3 | 20.1 | 74.1 | 20.3 | 58.0 | 26.9 | 64.2 | 26.4 | 66.9 | 19.8 | 63.0 | 29.7 |
| Ours | 54.2 | **14.1** | 68.7 | **12.7** | 78.6 | 15.8 | 51.1 | 14.8 | 61.0 | **8.8** | 68.0 | **10.5** | 58.3 | **9.2** |

| Method | $Pr \to Cl$ | | $Pr \to Rw$ | | $Rw \to Ar$ | | $Rw \to Cl$ | | $Rw \to Pr$ | | Average | |
|---|---|---|---|---|---|---|---|---|---|---|---|---|
| | OOD Acc.↑ | FPR↓ | OOD Acc.↑ | FPR↓ | OOD Acc.↑ | FPR↓ | OOD Acc.↑ | FPR↓ | OOD Acc.↑ | FPR↓ | OOD Acc.↑ | FPR↓ |
| STA$_{sum}$ [116] | 44.7 | 28.5 | 78.1 | 36.7 | **67.9** | 37.7 | 51.4 | 42.1 | 77.9 | 42.0 | 63.4 | 37.4 |
| STA$_{max}$ [116] | 44.2 | 32.9 | 76.2 | 35.7 | 67.5 | 33.3 | 49.9 | 38.9 | 77.1 | 44.6 | 61.8 | 36.7 |
| OSBP [113] | 44.5 | 33.7 | 76.2 | 28.3 | 66.1 | 32.7 | 48.0 | 37.0 | 76.3 | 31.4 | 64.1 | 33.7 |
| DAOD [117] | **58.4** | 57.2 | **81.8** | 49.4 | 66.7 | 56.7 | **60.0** | 63.4 | **84.1** | 65.3 | **69.6** | 49.8 |
| OSLPP [118] | 53.1 | 32.9 | 77.0 | 28.8 | 60.8 | 25.0 | 54.4 | 35.7 | 78.4 | 29.2 | 63.8 | 28.3 |
| ROS [119] | 46.5 | 28.8 | 70.8 | 21.6 | 67.0 | 29.2 | 51.5 | 27.0 | 72.0 | 20.0 | 61.6 | 27.6 |
| Anna [115] | 54.6 | 25.2 | 74.3 | 21.1 | 66.1 | 22.7 | 59.7 | 26.9 | 76.4 | 19.0 | 65.6 | 23.3 |
| Ours | 48.1 | **13.4** | 76.9 | **8.00** | 64.8 | **9.5** | 56.1 | **11.8** | 80.9 | **14.5** | 63.9 | **12.0** |

Table 5: Results on Office-Home. **Bold**=Best.

## E.4 Ablation Study

**Better adaptation to the heterogeneous distribution.** As presented in Table 6, the results underscore our competitive performance compared to state-of-the-art spectral learning approaches within their respective domains. For a fair comparison, SCL [7, 12] is purely unsupervised pre-trained on $\mathcal{D}_l \cup \mathcal{D}_u$, where $\mathcal{D}_l$ represents the labeled set, and $\mathcal{D}_u$ denotes the unlabeled wild set. NSCL [13] undergoes unsupervised pre-training on $D_u$ and supervised pre-training on $D_l$.

The improvement over SCL [7, 12] in both OOD generalization and detection illustrates the tremendous help given by labeled information, which also perfectly aligns with our theoretical insights in Appendx C. The comparison with NSCL [13] indicates that unsupervised pre-training on $\mathcal{D}_l \cup \mathcal{D}_u$ can contribute to the adaptation to the heterogeneous wild distribution, thereby establishing the generality of our method.

| $\mathbb{P}^{semantic}_{out}$ | Method | OOD Acc.↑ | ID Acc.↑ | FPR↓ | AUROC↑ |
|---|---|---|---|---|---|
| PLACES365 | SCL [7, 12] | 74.02 | 87.20 | 67.42 | 84.79 |
| | NSCL [13] | 86.79 | 91.56 | 54.27 | 87.07 |
| | Ours | **87.04** | **93.40** | **40.97** | **91.82** |
| LSUN-R | SCL [7, 12] | 63.77 | 84.86 | 4.10 | 99.29 |
| | NSCL [13] | 78.69 | 89.43 | 0.27 | 99.93 |
| | Ours | **79.68** | **92.44** | **0.06** | **99.99** |

Table 6: Comparison with spectral learning methods. We employ CIFAR-10 as $\mathbb{P}_{in}$ and CIFAR-10-C with Gaussian noise as $\mathbb{P}^{covariate}_{out}$. **Bold**=Best.

**Impact of semantic OOD data.** Table 7 empirically verifies the theoretical analysis in Section B. We follow Cao et al. [120] and separate classes in CIFAR-10 into 50% known and 50% unknown classes. To demonstrate the impacts of semantic OOD data on generalization, we simulate scenarios when semantic OOD shares the same or different domain as covariate OOD. Empirical results in Table 7 indicate that when semantic OOD shares the same domain as covariate OOD, it could significantly improve the performance of OOD generalization.

## F Implementation Details

**Training settings.** We conduct all the experiments in Pytorch, using NVIDIA GeForce RTX 2080Ti. We use SGD optimizer with weight decay 5e-4 and momentum 0.9 for all the experiments. In CIFAR-10 experiments, we pre-train Wide ResNet with spectral contrastive loss for 1000 epochs.

| Corruption Type of $\mathbb{P}_{out}^{covariate}$ | $\mathbb{P}_{out}^{semantic}$ | OOD Acc.↑ |
|---|---|---|
| Gaussian noise | SVHN | 85.48 |
| Gaussian noise | LSUN-C | 85.88 |
| Gaussian noise | Places365 | 83.28 |
| Gaussian noise | Textures | 86.84 |
| Gaussian noise | LSUN-R | 80.08 |
| Gaussian noise | Gaussian noise | **88.18** |

Table 7: The impact of semantic OOD data on generalization. Classes in CIFAR-10 are divided into 50% known and 50% unknown classes. The experiment in the last line uses known classes in CIFAR-10-C with Gaussian noise as $\mathbb{P}_{out}^{covariate}$ and novel classes in CIFAR-10-C with Gaussian noise as $\mathbb{P}_{out}^{semantic}$. **Bold**=best.

The learning rate (lr) is 0.030, batch size (bs) is 512. Then we use ID-labeled data to fine-tune for 20 epochs with lr 0.005 and bs 512. In ImageNet-100 experiments, we train ImageNet pre-trained ResNet-34 for 100 epochs. The lr is 0.01, bs is 512. Then we fine-tune for 10 epochs with lr 0.01 and bs 128. In Office-Home experiments, we use ImageNet pre-trained ResNet-50 with lr 0.001 and bs 64. We use the same data augmentation strategies as SimSiam [98]. We set K in KNN as 50 in CIFAR-10 experiments and 100 in ImageNet-100 experiments, which is consistent with Sun et al. [23]. And $\eta_u$ is selected within {1.00, 2.00} and $\eta_l$ is within {0.02, 0.10, 0.50, 1.00}. In Office-Home experiments, we set K as 5, $\eta_u$ as 3, and $\eta_l$ within {0.01, 0.05}. $\eta_u, \eta_l$ are summarized in Table 8.

| ID/Covariate OOD | Semantics OOD | $\eta_l$ | $\eta_u$ |
|---|---|---|---|
| CIFAR-10/CIFAR-10-C | SVHN | 0.50 | 2.00 |
| CIFAR-10/CIFAR-10-C | LSUN-C | 0.50 | 2.00 |
| CIFAR-10/CIFAR-10-C | Textures | 0.50 | 1.00 |
| CIFAR-10/CIFAR-10-C | Places365 | 0.50 | 2.00 |
| CIFAR-10/CIFAR-10-C | LSUN-R | 0.10 | 2.00 |
| ImageNet-100/ImageNet-100-C | iNaturalist | 0.10 | 2.00 |
| Office-Home Ar/Cl, Pr, Rw | Cl, Pr, Rw | 0.01 | 3.00 |
| Office-Home Cl/Ar, Pr, Rw | Ar, Pr, Rw | 0.01 | 3.00 |
| Office-Home Pr/Ar, Cl, Rw | Ar, Cl, Rw | 0.05 | 3.00 |
| Office-Home Rw/Ar, Cl, Pr | Ar, Cl, Pr | 0.05 | 3.00 |

Table 8: Selection of hyper-parameters $\eta_l, \eta_u$

**Validation strategy.** For validation, we could only access to unlabeled mixture of validation wild data and clean validation ID data, which is rigorously adhered to Scone [5]. Hyper-parameters are chosen based on the performance of ID Acc. on the ID validation set. We present the sweeping results in Table 9.

| $\eta_l$ | $\eta_u$ | ID Acc. (validation)↑ | ID Acc.↑ | OOD Acc.↑ | FPR↓ | AUROC↑ |
|---|---|---|---|---|---|---|
| 0.02 | 2.00 | 88.52 | 87.12 | 70.31 | 52.16 | 90.03 |
| 0.10 | 2.00 | 95.36 | 91.72 | 77.98 | 20.20 | 96.85 |
| 0.50 | 2.00 | 95.72 | 91.79 | 78.23 | 17.66 | 97.26 |
| 1.00 | 2.00 | 94.96 | 90.91 | 81.92 | 24.99 | 94.82 |
| 0.02 | 1.00 | 89.04 | 87.44 | 60.60 | 46.01 | 92.01 |
| 0.10 | 1.00 | 93.92 | 90.70 | 74.58 | 21.50 | 96.83 |
| 0.50 | 1.00 | **96.76** | 92.50 | 81.40 | 12.05 | 98.25 |
| 1.00 | 1.00 | 94.24 | 90.77 | 65.58 | 14.00 | 97.27 |

Table 9: Sensitivity analysis of hyper-parameters $\eta_l, \eta_u$. We employ CIFAR-10 as $\mathbb{P}_{in}$, CIFAR-10-C as $\mathbb{P}_{out}^{covariate}$, and Textures as $\mathbb{P}_{out}^{semantic}$. **Bold**=best.

