# OpenReview forum: "Bridging OOD Detection and Generalization: A Graph-Theoretic View"
_NeurIPS.cc/2024/Conference — NeurIPS 2024 poster_

### Official Review · Reviewer_LF4w · 2024-06-25

**Soundness:** 3
**Presentation:** 3
**Contribution:** 3
**Rating:** 6
**Confidence:** 3

**Summary:**

This work proposes a framework to address OOD detection and generalization jointly for image data, using a graph representation, where edges are constructed by both self-supervised data transformation probability and supervised labels. An example is provided with theoretical analysis, to show the advantages and disadvantages of the proposed method.

**Strengths:**

1. The paper is well-written with a clear motivation.

2. The paper proposes a algorithm with graph-based formulation to jointly address OOD detection and generalization, achieving state-of-the-art performance.

3. The paper presents an example with theoretical analysis to study the characteristics of the algorithm and better understand it.

4. The paper proposes a surrogate loss to enhance computational efficiency, which is well-supported.

**Weaknesses:**

1. From the theoretical analysis (lines 200-216), it appears that the OOD generalization ability of this algorithm depends on the relationship between $\alpha$ and $\beta$. This may lead to failures in OOD generalization, while for OOD detection, the method is more effective.

2. As the data transformation is not learnable, $\alpha$ and $\beta$ seem to be fully determined by the data distribution. Hence, there is no guarantee for OOD generalization in some scenarios.

3. The experimental evaluation is consistent with the theoretical characterization, indicating that the model is less effective at OOD and ID classification compared with its OOD detection ability.

4. Although the authors present a theoretical analysis, which reflects the advantages and disadvantages of this approach, however, for the OOD generalization failure and limited ID classification performance, the author doesn't propose further refinement to handle this issue.

**Questions:**

1. Would more advanced data transformation in SSL enhances the OOD generalization ability?

2. What does it mean by heterogeneous distribution in line 266? It seems the major difference to previous methods is the usage of both unsupervised and supervised learning when building the graph.

3. For the FPR metric, why can the proposed algorithm perform so much better than the state-of-the-art OOD detection method? For example, 0.13 vs. 40.76 in ASH in Table 1. Is there any explanation for this significant uplift?

---

> ### Author Rebuttal · Authors · 2024-08-04
>
> We thank you for the positive and constructive feedback on our work! Below we address your questions and comments in detail.
>
> > Further refinement for the OOD generalization performance
>
> We appreciate your insightful analysis and totally agree with your perspective. Guarantees for OOD generalization may not always hold in some specific scenarios. This is not unique to our approach, but a common challenge underlying OOD generalization algorithms (e.g., when the domain gap is significant) [1]. Inspired by your suggestion, a potential future direction could involve incorporating learnable augmentations, which may further enhance OOD generalization performance.
>
> > Would more advanced data transformation in SSL enhance the OOD generalization ability?
>
> This is an excellent question! In this paper, we chose to use the same data augmentation transformations as [2] to keep the method simple and user-friendly. We agree that exploring more advanced data transformations could be an interesting direction.
>
> > What does it mean by heterogeneous distribution in line 266?
>
> By "heterogeneous distribution," we refer to unlabeled data that includes various types of distributional shifts, such as covariate shifts and semantic shifts. This is formally defined in **Definition 2.1** in Section 2.
>
> In contrast, the baseline methods assume the unlabeled data exhibits a homogeneous shift, either entirely due to covariate shifts (as in unsupervised domain adaptation) or semantic shifts (as in novel class discovery).
>
> For greater clarity, we have revised the wording to "heterogeneous shift." Thanks for calling that out!
>
> > For the FPR metric, why can the proposed algorithm perform so much better than the state-of-the-art OOD detection method? For example, 0.13 vs. 40.76 in ASH in Table 1. Is there any explanation for this significant uplift?
>
> Great question. To clarify, ASH is a post-hoc OOD detection method, which operates on a model trained solely with ID data. In contrast, our method is trained on a combination of ID data and unlabeled data from $\mathbb{P}_\text{wild}$. OOD detection methods that use auxiliary data typically achieve significantly lower FPR compared to post-hoc methods. Therefore, a fairer comparison would be with methods trained on unlabeled data, such as Outlier Exposure, WOODS, and SCONE. Our method favorably outperforms these competitive baselines, including the state-of-the-art method SCONE [3].
>
> -----
> References
>
> [1] Ye, Haotian, et al. Towards a theoretical framework of out-of-distribution generalization. Advances in Neural Information Processing Systems 34 (2021): 23519-23531.
>
> [2] Chen, Xinlei, and Kaiming He. Exploring simple siamese representation learning. Proceedings of the IEEE/CVF conference on computer vision and pattern recognition. 2021.
>
> [3] Bai, Haoyue, et al. Feed two birds with one scone: Exploiting wild data for both out-of-distribution generalization and detection. International Conference on Machine Learning. PMLR, 2023.

---

> > ### Comment · Reviewer_LF4w · 2024-08-10
> >
> > Thanks for the authors' reply. I will keep positive score.

---

> > > ### Author Response · Authors · 2024-08-10
> > >
> > > Thank you for taking the time to read our response! We are glad to hear that our rebuttal addressed your concerns.

---

### Official Review · Reviewer_6JZj · 2024-07-05

**Soundness:** 3
**Presentation:** 3
**Contribution:** 3
**Rating:** 6
**Confidence:** 3

**Summary:**

The paper proposes a graph-theoretic framework to address out-of-distribution (OOD) generalization and detection. The framework models the data using a graph, where vertices represent data points and edges indicate similarities based on supervised and self-supervised signals. By leveraging spectral decomposition of the graph's adjacency matrix, the authors derive provable errors for OOD generalization and detection performance. Empirical results demonstrate that the effectiveness of the proposed approach.

**Strengths:**

- the connection between the OOD problem and partition/clustering analysis from graph theory is interesting

- the papers are well-written with examples for illustrations

**Weaknesses:**

- despite the case studies for the proposed frameworks/analysis, the proposed method in its current form seems to be impractical for large-scale problem

- the effectiveness of the analysis and proposed method heavily rely on how the edges are/should be constructed in the graph, which itself is a challenging and open question

- there are some connections between the analysis and claims that are not very strong (see questions below for more details)

- there are some minor formatting/presentation issues in line 192-195

**Questions:**

1. what are the advantages of doing OOD generalization and detection at the same time compared to have different method for each?

2. what are the novel insight for OOD from this graph theoretic perspective?

**Limitations:**

N.A

---

> ### Author Rebuttal · Authors · 2024-08-04
>
> We thank you for the positive and constructive feedback on our work! Below we address your questions and comments in detail.
>
> > Practicality of the algorithm/framework
>
> You raise an excellent point. Our graph-theoretic framework can be used practically, as detailed in Section 3.2. In particular, the spectral decomposition can be equivalently achieved by minimizing a surrogate objective in Equation (6), which can be efficiently optimized end-to-end using modern neural networks. **Empirically, we have demonstrated this on large-scale dataset including ImageNet (see Section E.2)**. Thus, our approach enjoys theoretical guarantees while being applicable to real-world data.
>
> > proposed method heavily rely on how the edges are/should be constructed in the graph.
>
> Thank you for raising this important point. We acknowledge that the construction of edges in a graph is indeed a challenging and open question, and this is a crucial aspect of our work.
>
> However, **our approach specifically addresses this challenge by proposing a surrogate objective that effectively circumvents the need for explicitly defining and constructing graph edges in a traditional sense**. Instead, we reformulate the problem into a contrastive learning framework, where the relationships between data points are implicitly captured through the use of augmentation transformations. This allows us to leverage the power of graph-based modeling while avoiding the complexities associated with direct edge construction.
>
> Moreover, our method is designed to be flexible and adaptable to different scenarios by allowing the augmentation transformation probabilities to guide the implicit graph structure. This design choice not only makes the approach more practical but also robust to variations in the underlying data distribution. By providing theoretical guarantees as a function of the parameters that define these probabilities, we ensure that our method is both effective and grounded in solid theoretical foundations. We believe our work represents a significant advancement in how graph-based methods can be applied to OOD detection and generalization.
>
>
>
> > What are the advantages of doing OOD generalization and detection at the same time compared to have different method for each?
>
> There are several benefits of devising a method that can jointly handle both OOD generalization and detection problems:
>
> - **Computational efficiency**: Investing in developing and maintaining a single method is typically more cost-effective than supporting two separate methods in inference time. This can be particularly important for organizations with limited resources, or dealing with large volume of data traffic.
>
> - **Deployment simplicity**: Deploying and maintaining a single model is generally simpler and less error-prone than managing multiple models. This includes considerations like updates, scaling, and monitoring.
>
> - **Improved performance**: The learning tasks of OOD generalization and detection can benefit from each other, as we have demonstrated in this paper. As shown in Table 1, our method excels in both OOD detection and generalization performance, surpassing state-of-the-art methods by a large margin.
>
> > Novel insights
>
> There are several theoretic insights derived from the graph-theoretic perspective:
>
> + **Effectiveness of OOD detection**: As presented in Section 4.3,  we primarily analyze the OOD performance in our proposed framework. Theorem 4.2 exhibits that the separability between semantic OOD data and ID data displays a large value, which facilitates OOD detection. The empirical results in Table 1 also validate the effectiveness of OOD detection performance.
>
> + **Novel analysis of semantic OOD data**: As presented in Section B, we introduce a novel analysis of the impact of semantic OOD data, thoroughly examining cases where semantic OOD data originates from the same or different domain as covariate OOD data. Theorem B.1. demonstrates that semantic OOD and covariate OOD sharing the same domain could benefit OOD generalization, which can be empirically validated in Section E.4.
>
> + **Impact of ID labels on OOD performance**: As presented in Section C, we investigate the effects of ID labels on OOD generalization and detection, providing new insights into how the linear probing accuracy of covariate OOD and separability between ID and semantic OOD data improve with the incorporation of ID label information. Theorem C.1. and C.2. show that incorporating ID labels during pre-training can facilitate both OOD generalization and detection performance, which can also be validated empirically in Tables 2 and 6.
>
> > Formatting issue in L192-L195
>
> Great catch! We will fix that in the revised version.

---

> > ### Comment · Reviewer_6JZj · 2024-08-10
> >
> > Thank you for the response. I do not have further questions.

---

> > > ### Author Response · Authors · 2024-08-10
> > > **Reply**
> > >
> > > Thank you for taking the time to read our response! We are glad to hear that our rebuttal addressed your concerns.

---

### Official Review · Reviewer_ureZ · 2024-07-08

**Soundness:** 3
**Presentation:** 3
**Contribution:** 2
**Rating:** 6
**Confidence:** 4

**Summary:**

The paper introduces a novel graph-theoretic framework to tackle both out-of-distribution (OOD) generalization and detection. By representing data points as vertices in a graph and using the adjacency matrix decomposition, the authors derive data representations that allow for quantifiable error rates in OOD tasks. Theoretical insights are provided through formal error bounds, and empirical results demonstrate the framework's effectiveness and robustness, showcasing significant improvements over existing methods. This framework is practical and scalable, leveraging modern neural networks for efficient optimization on real-world data.

**Strengths:**

1. The idea is novel and the model is both reasonable and sound.
2. The intuition behind the method is clearly described, and the theoretical analysis justifies the model well.
3. Comprehensive experiments demonstrate the effectiveness and robustness of the proposed method, with significant improvements over state-of-the-art techniques.
4.  The framework is scalable and practical, utilizing modern neural networks for efficient optimization, making it applicable to real-world data.
 5. The use of spectral decomposition of the graph’s adjacency matrix to derive data representations is a novel and effective approach.

**Weaknesses:**

1.The paper builds on the baseline Scone for its experiments. Regarding Definition 2.2, it is unclear why four metrics are used in the experiments while only three are introduced here. Is it because Scone only introduced three? Please provide a reasonable explanation.

2.The field of Graph Neural Networks (GNNs) has mature tasks for OOD generalization and OOD detection. Why did the authors choose to use graphs to address tasks in computer vision instead of leveraging these existing GNN-specific tasks?

3.While the authors provide extensive theoretical insights and proofs, the experimental section seems insufficient. The main experiments are largely based on Scone, and the additional experiments only compare performance. If this work is pioneering, this might be acceptable, but given related prior work, the authors should not limit themselves to performance comparisons. They should demonstrate the advantages of their approach in other dimensions, such as time and space efficiency, to prove its superiority over Scone.

4.Figure 4 only visualizes the method proposed in this paper, lacking comparison with other methods.

5.The theoretical analysis in this paper is based on certain assumptions that may not hold in all practical situations, potentially limiting the applicability of the results. These assumptions include the accuracy and representativeness of the graph representations and spectral decompositions used to quantify OOD generalization and detection errors, as well as the ideal conditions for spectral decomposition and the relationships between ID and OOD data. Additionally, the analysis assumes specific distributions of wild data, linear separability of OOD data in the learned representation space, and the particular structure of the graph used in the analysis.

**Questions:**

1.The theoretical analysis relies on several assumptions, such as the accuracy of graph representations and spectral decompositions, and the linear separability of OOD data. Can the authors provide more details on these assumptions and discuss the potential impact if these assumptions do not hold in practical scenarios?

2.The paper uses graphs to handle tasks in computer vision, despite the existence of well-established OOD generalization and detection tasks within the domain of Graph Neural Networks. What motivated the authors to choose this approach? Are there specific advantages that this graph-theoretic framework provides for computer vision tasks?

---

> ### Author Rebuttal · Authors · 2024-08-04
>
> We thank you for the positive and constructive feedback! Below we address your questions and comments in detail.
>
> > Questions about the metrics
>
> We are happy to clarify this. SCONE adopted four metrics in experiments, **which are identical to ours**. Following SCONE, we introduce the FPR as the primary metric for evaluating OOD detection in problem setup. Since both AUROC and FPR are commonly used, we also report AUROC in our experiments for comprehensive evaluation. For better clarity, we will include AUROC in Definition 2.1.
>
> > GNN-specific tasks
>
> Thank you for highlighting the connection to the GNN literature! We chose to focus on computer vision tasks for several key reasons:
>
> 1. **Established framework**: Image classification tasks are well-established and have been extensively studied in the OOD detection and generalization literature. This allows us to position our work within a widely recognized and classical setting.
>
>
> 2. **Consistency with prior work**: Our problem setup closely builds on prior work, particularly SCONE, which also focuses on the image domain. To ensure evaluation consistency and fair comparisons, it was important for us to remain within this established framework.
>
>
> 3. **Novel graph-based perspective**: Our work introduces significant insights and techniques by applying a graph-based perspective to computer vision tasks. Unlike traditional GNN-based tasks, constructing edges and graphs for image-based tasks is less straightforward and insufficiently explored. To address this, a key innovation of our paper is the introduction of a surrogate objective, which reformulates the graph factorization problem as a contrastive learning objective. This allows our practical algorithm to be efficiently optimized without explicitly operating on a graph adjacency matrix, while still benefiting from the theoretical guarantees provided by the underlying graph-theoretic formulation.
>
> > Clarification on experiments
>
> We believe our experiments are comprehensive and offer significant advancements over SCONE.
>
> 1. We conduct more extensive evaluations than SCONE. In Section E.2, we present results on the ImageNet-100 dataset, and in Section E.3, on the Office-Home dataset—which was not considered by SCONE. These results consistently demonstrate our superior performance compared to SCONE.
>
>
> 2. We introduce a novel analysis and ablation study on the impact of semantic OOD data, thoroughly examining cases where semantic OOD data originates from the same or different domain as covariate OOD data. This analysis is theoretically explored in Section B and empirically validated in Section E.4, a focus that SCONE does not address.
>
> 3. We investigate the effects of ID labels on OOD generalization and detection, providing new insights into how the performance improves with the incorporation of ID label information (Section C). The empirical results shown in Tables 2 and 6 also validate our theoretical insights.
>
> Lastly, we would like to clarify that the inference time and space efficiency of our method are identical to those of SCONE, as both approaches utilize the same neural network backbone. Overall, our work represents a significant theoretical and empirical advancement over SCONE.
>
> > More visualizations
>
> As suggested, we present visualizations of SCL in the attached PDF. Compared with the baseline, our learning framework effectively pushes the semantic OOD data to be apart from the ID data and pulls the covariate OOD data close to the ID data in the embedding space. Visualization of SCONE can be found in Figure 3 of their paper.
>
> > Theoretical assumptions
>
> Thanks for the thoughtful question. We are happy to clarify this further.
>
> 1. _Accuracy of graph representations and spectral decompositions_: Our theoretical framework indeed leverages graph representations and spectral decompositions, specifically relying on the structure induced by augmentation transformations. However, we want to emphasize that our method does not depend on perfect or exact graph representations. Instead, our approach is designed to be robust to variations in the graph structure by incorporating empirical augmentation probabilities that guide the construction of the graph. The spectral decompositions used in our analysis are derived in exact closed form, following standard singular value decomposition, without requiring any additional assumptions.
>
> 2. _Linear separability of OOD data_: Contrary to the concern raised, **our theoretical analysis does not assume the linear separability of OOD data**. In fact, one of the strengths of our approach is its ability to handle cases where linear separability is not guaranteed. Our theorems explicitly account for scenarios where the linear probing error and ID-OOD separability are non-zero. This reflects real-world conditions where OOD data might not be linearly separable, yet our method can still provide meaningful guarantees and effective performance.
>
> 4. _Impact of assumptions in practical scenarios_: In practical applications, it is indeed possible that some of the idealized conditions assumed in theoretical analysis might not fully hold. However, our method is designed with flexibility in mind. For example, the parameters ($\alpha$, $\rho$, $\gamma$ and $\beta$) governing the augmentation transformation probabilities are designed with generality to capture different practical scenarios. **Our guarantees are provided as a function of these parameters, which can be flexibly adjusted to fit the specific characteristics of a given dataset**. Additionally, our empirical results demonstrate that the method performs robustly across a range of datasets and conditions, indicating that the assumptions made do not overly constrain the practical applicability of our approach. We believe this balance between theoretical rigor and practical flexibility is a key strength of our work.
>
> We will revise our manuscript to make these points clear - thank you again for your valuable comments!

---

> > ### Comment · Reviewer_ureZ · 2024-08-10
> >
> > Thank you for addressing the concerns I raised in my previous review. I have decided to increase my score for this submission.

---

> > > ### Author Response · Authors · 2024-08-10
> > >
> > > Thank you for taking the time to read our response and for increasing the score! We are glad to hear that our rebuttal addressed your concerns.

---

### Official Review · Reviewer_DU8N · 2024-07-18

**Soundness:** 2
**Presentation:** 3
**Contribution:** 2
**Rating:** 5
**Confidence:** 3

**Summary:**

This paper proposes a unified framework for OOD detection and generalization, which first constructs a graph that includes both labeled and unlabeled data and derives data representations by factorizing the graph’s adjacency matrix. These representations help quantify OOD generalization and detection performance. The framework's effectiveness is demonstrated through experiments on CIFAR-10, ImageNet, and Office-Home datasets.

**Strengths:**

1. The authors discuss a new method for generalizing to covariate shifts while robustly detecting semantic shifts, providing valuable insights into OOD problem.

2. In addition to mathematical equations, the paper includes an illustrative example to clarify the method, enhancing understanding.

3. The proposed method achieves better performance in OOD generalization and detection compared to the baseline methods.

**Weaknesses:**

1. The effectiveness of the methods is limited by numerous hyperparameter selections, preventing its practical application in real-world scenarios.

2. The process for determining the distribution of augmentation $\tau$, which is the key points for constructing the graph, is not clear. he appropriateness and effectiveness of the formulation in equation (9) for various datasets and other cases remain uncertain.

3. The theoretical guarantees discussed in the paper are not clearly defined, making it unclear what the specific goals of these guarantees are.

**Questions:**

Please refer to the weaknesses.

---

> ### Author Rebuttal · Authors · 2024-08-04
>
> We thank the reviewer for the constructive feedback. We are glad to hear that you found our paper insightful, clearly presented, and performing well. We address each of your concerns below in detail:
>
> > Clarification on the number of hyperparameters
>
> Thank you for bringing up this point. To clarify, the practical algorithm is presented in Equation (6), which reformulates the graph factorization problem into a contrastive learning objective that can be efficiently trained end-to-end using a neural network.  **Importantly, there are only two hyperparameters involved in our learning objective: $\eta_l$ and $\eta_u$**. This is comparable to, or even simpler than, many existing methods in the field, which often involve multiple hyperparameters across different components of their models.
>
> To further assist in practical application, we provide guidelines and default values for the key hyperparameters in our paper. These guidelines (Section F) can serve as a starting point, reducing the burden of hyperparameter tuning in new scenarios. Additionally, these two hyperparameters have intuitive interpretations (e.g., balancing the influence of labeled vs. unlabeled data), which can help practitioners make informed adjustments based on the specific characteristics of their data.
>
>
> > Clarification on augmentation transformation in Equation (9)
>
> **In practice, there is no need to determine manually the augmentation transformation probability in $\mathcal{T}(x|\bar x)$**. Our practical algorithm, as described in Section 3.2, does not rely on the explicit construction of the graph or the augmentation transformation probability, making it adaptable to different datasets. **Specifically, the graph decomposition can be equivalently achieved by minimizing a surrogate contrastive learning objective**, which operates on pairs of images. In objective (6), empirical samples of augmented images (using common augmentations [1] such as Gaussian blur, color distortion, and random cropping) are sufficient for optimization, eliminating the need to know the underlying distribution of $\mathcal{T}(\cdot | \bar x)$. This not only makes the approach more practical but also adaptable to various datasets and use cases.
>
> We explicate the augmentation transformation probability primarily to support the theoretical analysis and provide tractable guarantees on how they impact OOD generalization and OOD detection performance. By providing theoretical guarantees as a function of the parameters that define these probabilities, we ensure that our method is grounded in solid theoretical foundations. The validity of the formulation in Equation (9) can also be supported in prior work [2]. **Thus, our approach enjoys theoretical guarantees while being easily applicable to various real-world datasets, as we have shown in our extensive experiments.**
>
> [1] Ting Chen, Simon Kornblith, Mohammad Norouzi, and Geoffrey E. Hinton. A simple framework for contrastive learning of visual representations. In ICML, volume 119 of Proceedings of Machine Learning Research, pages 1597–1607. PMLR, 2020.
>
> [2] Kendrick Shen, Robbie M. Jones, Ananya Kumar, Sang Michael Xie, Jeff Z. HaoChen, Tengyu Ma, and Percy Liang. Connect, not collapse: Explaining contrastive learning for unsupervised domain adaptation. In ICML, pages 19847–19878. PMLR, 2022.
>
> > Goals of theoretical guarantees
>
> **The goals and targets of our theoretical analysis are explicitly defined in Section 4.2**. Specifically,
> - We use linear probing evaluation (Equation 7) to quantify OOD generalization performance, measuring the misclassification rate on covariate-shifted OOD data.
> - We use separability evaluation (Equation 8) to quantify OOD detection performance.
>
> _Our Theorem 4.1 and Theorem 4.2 provide closed-form guarantees on these two errors respectively_. We are happy to revise the manuscript and highlight this connection more clearly. Thanks again for your valuable comments and suggestions!

---

> > ### Author Response · Authors · 2024-08-11
> >
> > Dear reviewer DU8N,
> >
> > We wanted to touch base with you as the deadline for the author-reviewer discussion phase is approaching soon. We trust you've had the opportunity to review our rebuttal, and we would be more than happy to address any further concerns you have.
> >
> > Thank you once again for your time and dedication to this review process. We look forward to your response and to furthering the dialogue on our manuscript.
> >
> > Best,
> >
> > Authors

---

> > ### Comment · Reviewer_DU8N · 2024-08-12
> >
> > Thanks for addressing my concerns. I would like to raise my score.

---

> > > ### Author Response · Authors · 2024-08-12
> > >
> > > Thank you for taking the time to read our response and for increasing the score! We are glad to hear that our rebuttal addressed your concerns.

---

### Author Rebuttal · Authors · 2024-08-05

We thank all the reviewers for their time and commitment to providing valuable feedback and suggestions on our work. We are encouraged that reviewers find our idea to be **novel**,  **interesting**, and **effective** (DU8N, ureZ, 6JZj), that our theoretical insights are **sound and valuable** (DU8N, ureZ), and that our results are **comprehensive and significant** (ureZ, LF4w). We are also encouraged that reviewers recognize our method to be **scalable**, **computationally efficient** and **practical** for real-world data (ureZ, LF4w). Additionally, we appreciate the acknowledgment of our **clear writing and presentation** (DU8N, 6JZj, LF4w).

As recognized by multiple reviewers, the significance of our work can be summarized as follows:

- Our work offers a new algorithmic framework that leverages graph-theoretic formulation to jointly address OOD detection and generalization problems, which is more challenging than addressing either problem alone.
- The framework is grounded in the spectral decomposition of a graph, which can be equivalently realized by minimizing a surrogate contrastive learning objective. This approach enhances the computational efficiency and practicality of our framework.
- Our framework provides theoretical guarantees while demonstrating effectiveness across various real-world datasets. We include sufficient ablations and illustrative examples to aid readers in understanding our method.


We respond to each reviewer's comments in detail below. The PDF attached includes the visualization requested by Reviewer ureZ. In response to the valuable suggestions provided by the reviewers, we will further refine our manuscript to clarify aspects that could benefit from additional explanation.

---

### Decision · Program_Chairs · 2024-09-25

**Decision:**

Accept (poster)

**Comment:**

All four reviewers expressed positive support to this work, which is considered to be theoretically sound and useful. Thus an accept is recommended.